https://doi.org/10.1038/s41467-020-19490-6　　**OPEN**

# Lkb1 suppresses amino acid-driven gluconeogenesis in the liver

Pierre-Alexandre Just[1,2,8], Sara Charawi[1,8], Raphaël G. P. Denis [3], Mathilde Savall[1], Massiré Traore[1], Marc Foretz [1], Sultan Bastu[1], Salimata Magassa[4], Nadia Senni[1], Pierre Sohier[1], Maud Wursmer[1], Mireille Vasseur-Cognet[5], Alain Schmitt[1,6], Morgane Le Gall[1,7], Marjorie Leduc[1,7], François Guillonneau [1,7], Jean-Pascal De Bandt[4], Patrick Mayeux[1,7], Béatrice Romagnolo[1], Serge Luquet[3], Pascale Bossard[1] & Christine Perret [1✉]

Excessive glucose production by the liver is a key factor in the hyperglycemia observed in type 2 diabetes mellitus (T2DM). Here, we highlight a novel role of liver kinase B1 (Lkb1) in this regulation. We show that mice with a hepatocyte-specific deletion of *Lkb1* have higher levels of hepatic amino acid catabolism, driving gluconeogenesis. This effect is observed during both fasting and the postprandial period, identifying Lkb1 as a critical suppressor of postprandial hepatic gluconeogenesis. Hepatic *Lkb1* deletion is associated with major changes in whole-body metabolism, leading to a lower lean body mass and, in the longer term, sarcopenia and cachexia, as a consequence of the diversion of amino acids to liver metabolism at the expense of muscle. Using genetic, proteomic and pharmacological approaches, we identify the aminotransferases and specifically Agxt as effectors of the suppressor function of Lkb1 in amino acid-driven gluconeogenesis.

[1] Université de Paris, Institut Cochin, INSERM, CNRS, F75014 Paris, France. [2] APHP, Centre–Université de Paris, Paris, France. [3] Unité de Biologie Fonctionnelle et Adaptative, Centre National la Recherche Scientifique, Unité Mixte de Recherche 8251, Université Paris Diderot, Sorbonne Paris Cité, 75205 Paris, France. [4] EA4466, PRETRAM, Université Paris Descartes, Paris, France. [5] UMR IRD 242, UPEC, CNRS 7618, UPMC 113, INRA 1392, Sorbonne Universités Paris and Institut d'Ecologie et des Sciences de l'Environnement de Paris, Bondy, France. [6] Electron Miscroscopy Facility, Institut Cochin, F75014 Paris, France. [7] 3P5 proteom'IC Facility, Université de Paris, Institut Cochin, INSERM, CNRS, F-75014 Paris, France. [8] These authors contributed equally: Pierre-Alexandre Just, Sara Charawi. ✉email: christine.perret@inserm.fr

The liver plays a major role in maintaining normal glycemia, by regulating the processes of glycogen breakdown (glycogenolysis), glycogen synthesis (glycogenogenesis) and glucose synthesis (gluconeogenesis) to produce glucose during fasting and to store glucose in the postprandial period. During fasting, glucose is initially generated by glycogenolysis, and then by gluconeogenesis from various gluconeogenic precursors, including amino acids. The liver is the principal organ capable of glucose synthesis[1]. During fasting, glucose is initially generated by glycogenolysis, and then by gluconeogenesis from various gluconeogenic precursors, including amino acids. However, hepatic gluconeogenesis is not restricted to the fasted state; it is also a normal physiological event in the fed state, as part of the metabolism of excess dietary amino acids, the carbon moiety of which is stored as glycogen, via the indirect pathway of glycogen synthesis[2,3]. A deregulation of gluconeogenesis is a signature of type 2 diabetes mellitus (T2DM). A deeper understanding of the mechanisms regulating hepatic gluconeogenesis is crucial for advances in the management of T2DM[4–6].

The liver kinase B1 (LKB1) pathway is known to suppress hepatic gluconeogenesis[7–10]. LKB1 is a master upstream kinase that directly phosphorylates and activates AMP-activated protein kinase (AMPK) and 12 kinases related to AMPK, known as AMPK-related kinases (ARKs), including the salt-inducible kinases (SIKs)[11]. AMPK is a key intracellular energy sensor and regulator of multiple metabolic processes[12]. Shaw et al. showed that the loss of Lkb1 in the liver results in hyperglycemia, due to an enhancement of gluconeogenesis, and identified Ampk as the enzyme responsible for induction of the gluconeogenic program by Lkb1 deficiency[7]. However, subsequent studies called the role of Ampk into question, instead identifying Siks as the effectors of gluconeogenesis suppression by Lkb1 in the liver[8,10,13].

Here, using mice with a hepatocyte-specific Lkb1 deficiency, we characterize a new role of Lkb1 in the control of hepatic amino acid catabolism, identifying Lkb1 as a suppressor of gluconeogenesis from amino acids. We show that, in addition to its known effects on hepatic gluconeogenesis, the loss of hepatic Lkb1 increases the uptake and catabolism of amino acids in the liver and, ultimately, their utilization as substrates for gluconeogenesis. This is the case not only during fasting, but also during the postprandial period, identifying Lkb1 as a suppressor of postprandial gluconeogenesis. The increase in hepatic amino acid extraction in mutant mice is associated with a decrease in plasma amino acid concentration, ultimately affecting protein homeostasis in the skeletal muscle. Indeed, mutant mice have a lower lean body mass and muscle amino acid content than wild-type mice. In the longer term, the impairment of amino acid metabolism leads to the development of sarcopenia and cachexia, causing premature death in more than 60% of the mutant mice. Using genetic, proteomic, and pharmacological approaches, we identify aminotransferases and, specifically, Agxt as effectors, of the suppressor function of Lkb1 in amino acid-driven gluconeogenesis. We find that this effect of Lkb1 occurs independently of Ampk and identify a phosphorylated RNA-binding protein network controlled by Lkb1 as the key element in controlling the expression of enzyme of amino acid catabolism at the translational level.

## Results

### Hepatic loss of Lkb1 induced hyperglycemia and sarcopenia.
Tamoxifen was injected into adult Lkb1fl,fl;TTR-CreTam and control Lkb1fl,fl mice to induce the deletion of Lkb1 specifically in the hepatocytes. The mutant mice are referred to hereafter as Lkb1KOlivad mice. Their phenotypes were analyzed 15 days later, to characterize the immediate phenotype resulting from Lkb1

deletion in the liver. Western blots confirmed the absence of Lkb1 in the livers of Lkb1KOlivad mice. As expected, Ampk phosphorylation levels were much lower in the livers of mutant mice than in those of controls (Fig. 1a).

Tamoxifen-treated Lkb1KOlivad mutant mice presented hyperglycemia in the fasted state and were glucose-intolerant in intraperitoneal (IP) glucose tolerance tests, consistent with the findings of a previous study[7] (Fig. 1b, Supplementary Fig. 1a). Mutant mice also displayed hyperglycemia in the refed state (Fig. 1b). Plasma insulin concentrations did not differ significantly between mutant mice and controls, for any of the nutritional states considered (Fig. 1c). No significant differences in plasma and hepatic triglyceride levels were observed between mutant and wild-type mice (Supplementary Fig. 1b, c). Mutant mice had slightly, but significantly heavier livers (Fig. 1d) with preserved liver function (Supplementary Fig. 1d, e) and a normal architecture on liver histology examination (Supplementary Fig. 1g). However, the PAS staining of liver sections associated with the determination of glycogen content revealed a much higher level of glycogen accumulation in mutant than in control mice, in the fasted state (Fig. 1e). We investigated this increase in glycogen content in Lkb1KOlivad mice, by studying two critical enzymes involved in the control of glycogen storage: glycogen synthase (Gys2), which is involved in glycogen synthesis, and glycogen phosphorylase (Pygl), which is involved in glycogen degradation[6]. During the fasting period, the active form of Pygl (pS15Pygl) was much less abundant in Lkb1KOlivad mice than in controls, suggesting a much lower levels of glycogen degradation; at the same time levels of the Gys2 protein were significantly higher (Fig. 1f). Given the role of glucagon as a counter-regulator of insulin signaling for glucose homeostasis, we checked its plasma levels in the fasting and refed state and did not find any significant difference in the insulin/glucagon ratio, suggesting that a deregulation of glucagon secretion was not involved in the phenotype of the Lkb1KOlivad mice (Supplementary Fig. 1f). Together, these findings can account for the higher glycogen content of the livers of fasted Lkb1KOlivad mice.

We also monitored Lkb1KOlivad mice for several months, to analyze their long-term phenotypes. About 6 months after tamoxifen injection, more than 60% of the mutant mice died from severe sarcopenia and cachexia (Fig. 1g). Mutant mice had a much lower body weight (Fig. 1i), with a striking lack of skeletal muscle and adipose tissue (Fig. 1h–j). Tibialis and gastrocnemius muscle weights were strongly decreased (Fig. 1j), and morphometric analysis showed that sarcopenia was linked to a smaller muscle fiber size (Supplementary Fig. 1h).

Overall, our results show that the specific deletion of Lkb1 in hepatocytes leads to the impairment of glucose homeostasis, with hyperglycemia and an abnormal accumulation of glycogen during the fasting period. In the long term, the hepatocyte-specific inactivation of Lkb1 led to severe cachexia and sarcopenia, resulting in death in more than 60% of the animals.

### Body composition of mutant mice reveals lower lean body mass.
The cachectic and sarcopenic phenotype developing in Lkb1KOlivad mice over time clearly suggests that hepatic Lkb1 loss is responsible for changes in the relationships between the liver and other organs, such as muscles. We monitored changes in body composition in Lkb1KOlivad and control mice by MRI for nine weeks after tamoxifen injection, to confirm these peripheral consequences of liver-specific Lkb1 loss. No significant difference was found between mutant and control mice in terms of overall body weight gain, but the mutant mice displayed significant modifications of their body composition, with a lower lean mass

and a higher fat mass (Fig. 2a–c). The monitoring of mice in indirect calorimetry cages revealed no significant differences in food intake or total energy expenditure between LKBKO[livad] and control mice, and significantly lower levels of locomotor activity during the night in LKBKO[livad] mice than in their control littermates, consistent with a positive energy balance (Fig. 2d–g). In addition, mutant mice displayed a whole-body shift in substrate preference toward higher levels of carbohydrate oxidation.

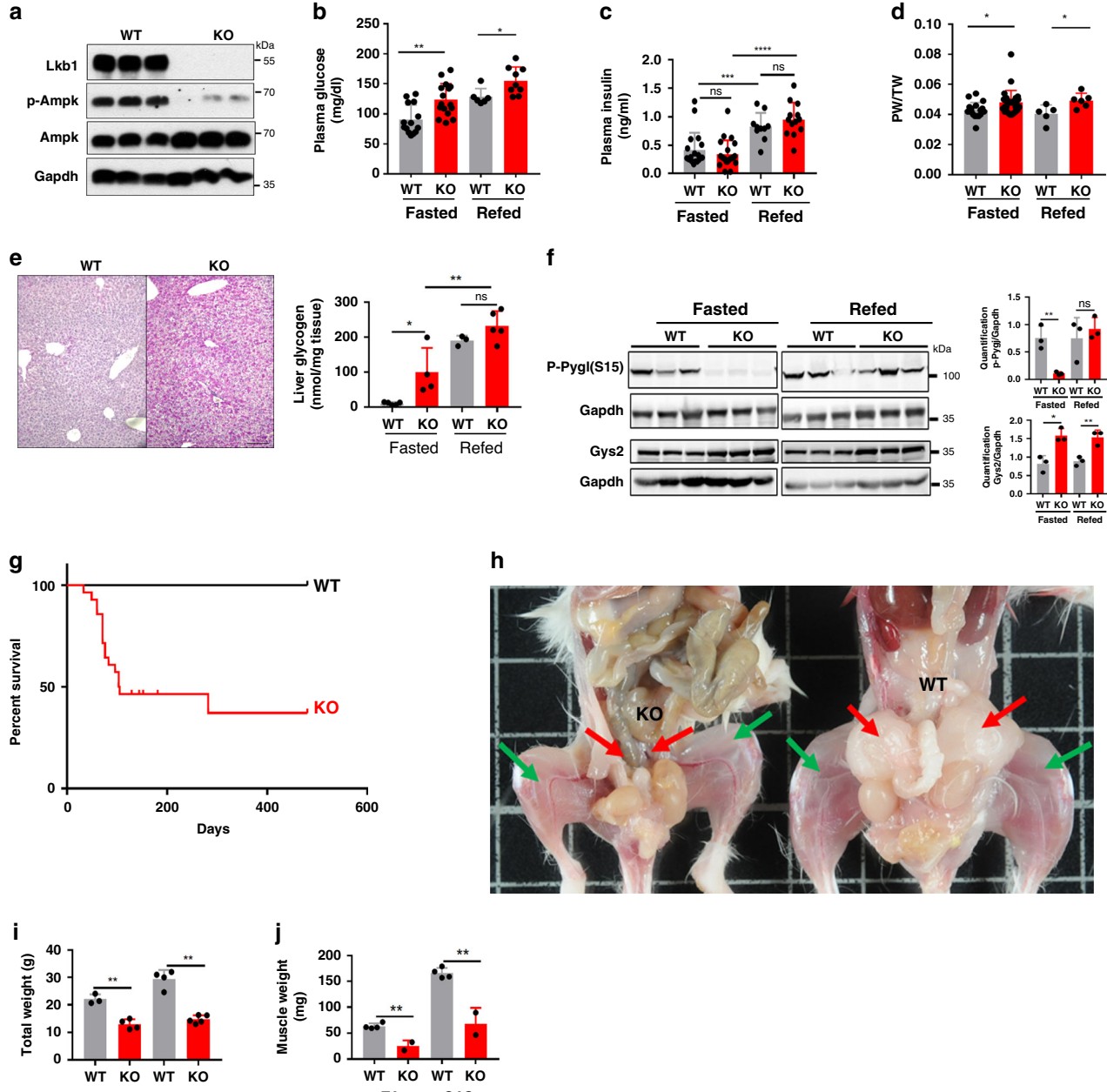

**Fig. 1 Short- and long-term phenotypes of Lkb1KO[livad] mice. a** Immunoblot showing the efficient inactivation of *Lkb1* in the livers of Lkb1KO[livad] (KO, $n = 3$) mice and lower levels of Ampk phosphorylation compared to control littermate (WT, $n = 3$). β-actin is the loading control. Representative blot of four independent experiments. **b–f** Short-term phenotype, with impaired glucose homeostasis. Animals were analyzed 15 days after the injection of tamoxifen. **b** Blood glucose levels of fasted mice (WT: $n = 14$, KO: $n = 16$) and refed mice (KO: $n = 15$, WT: $n = 16$). **c** Plasma insulin concentration in fasted (WT: $n = 14$, KO: $n = 16$) and refed (WT: $n = 10$, KO: $n = 13$) mice. **d** Ratio of liver weight to total body weight (LW/TW) in fasted (WT: $n = 14$, KO: $n = 16$) and refed (WT: $n = 5$, KO: $n = 5$) mice. **e** Periodic acid Schiff staining of sections of the liver from fasted mice (scale bar: 100 μm), and glycogen content in fasted (WT: $n = 8$, KO: $n = 14$) and refed (WT: $n = 3$, KO: $n = 3$) mice. **f** Immunoblot of Pygl phosphorylation (pS15) and Gys2 levels in fasted and refed mice. Quantification was performed with FUJI multigauge software and normalized to Gapdh, $n = 3$ mice per group. Representative blot of two independent experiments. **g–j** Long-term phenotype, with sarcopenia and cachexia. **g** Kaplan–Meier survival curves of mutant (KO, $n = 29$) and control (WT, $n = 10$) mice. **h** Gross morphology of a control littermate (WT) and a cachexic mutant (KO) showing a lack of fat mass (red arrow) and muscle sarcopenia (green arrow) in the mutant mice. **i** Total weight of female (KO: $n = 4$) and male (KO: $n = 5$) mutant mice at the time of euthanasia due to cachexia, relative to those of their control littermates (WT, female: $n = 3$, male: $n = 4$). **j** Weight of tibialis (TA) and gastrocnemius (GAS) muscles of male mutant mice (KO, $n = 3$) at the time of euthanasia due to cachexia, relative to those of their control littermates (WT, $n = 4$). All graphical data are means values ± SD. P values were determined by unpaired two-tailed *t*-test. *$p \leq 0.05$; **$p \leq 0.01$. Source data are provided as a Source data file.

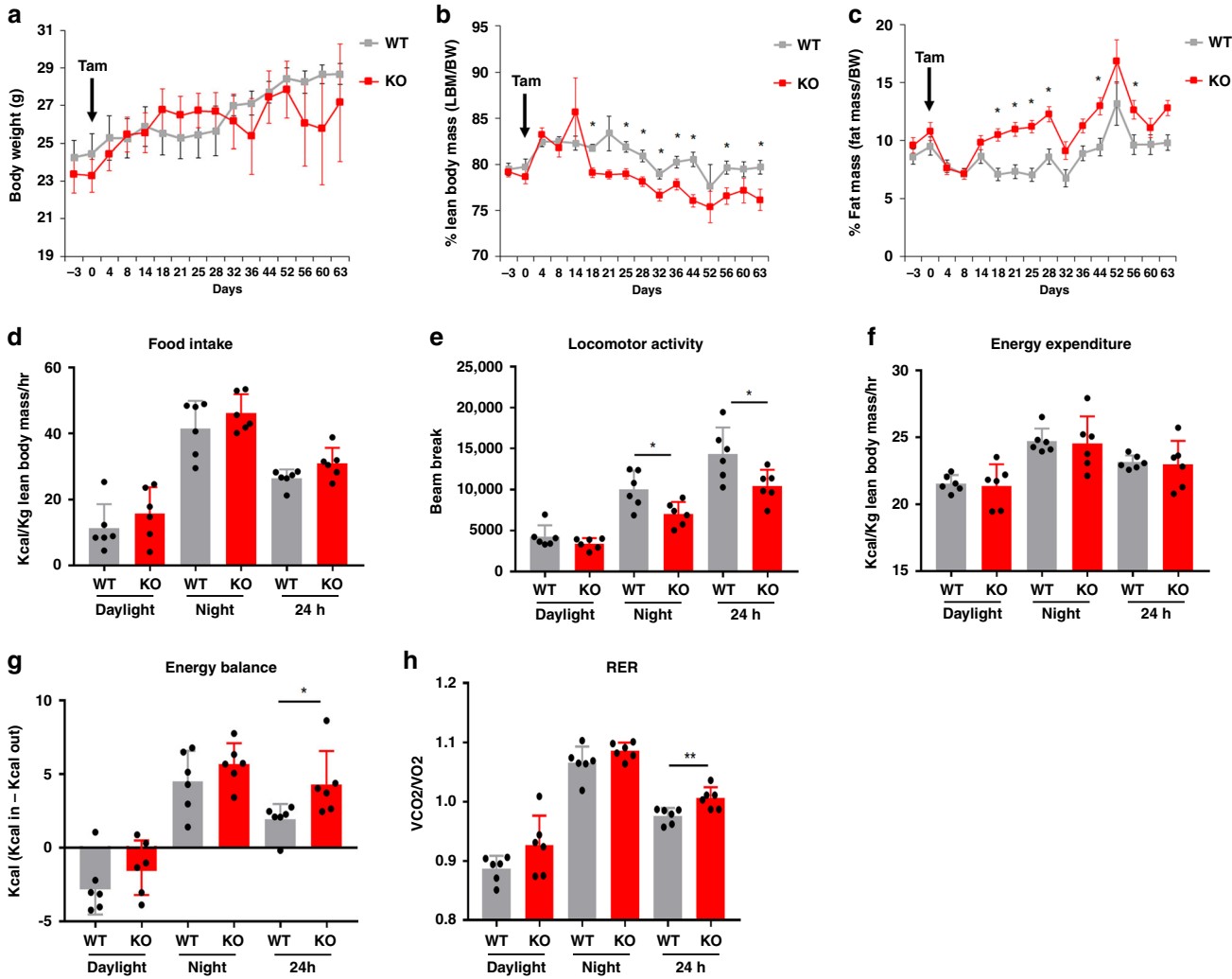

**Fig. 2 *Lkb1* deletion in the hepatocytes affects body composition and whole-body metabolism. a–c** Changes in whole-body composition analyzed after tamoxifen injection in Lkb1KO^livad (KO) mice. **a** Body weight (g) and of the percentage of total body weight accounted for by lean body mass (**b**) (g/bw) and fat mass (**c**) (g/bw), for KO (*n* = 6) and control (WT, *n* = 6) mice over a period of 9 weeks. Data are expressed as means ± SEM. *P* values were determined by unpaired two-tailed *t*-test. *$p \leq 0.05$. Tamoxifen was injected into the animals on day 0 (black arrow); calorimetric analysis was performed from day 14 to day 18. **d** Mean daylight-, night- and 24 h mean food intake (g/kg lean body mass/h). **e** Mean daylight-, night- and 24 h- mean spontaneous locomotor activity (beam break). **f** Mean daylight-, night- and 24 h mean energy expenditure (kcal/kg lean body weight/h). **g** Mean daylight-, night- and 24 h mean energy balance. **h** Mean daylight-, night- and 24 h mean respiratory exchange ratio (VCO₂/VO₂) of KO and control mice. Means over 24 h were calculated from 96 h measurements. All graphical data (**d–h**) are expressed as the mean ± SEM of 6 animals per group. *P* values were determined by unpaired two-tailed *t*-test. *$p \leq 0.05$; **$p \leq 0.01$. Source data are provided as a Source data file.

Indeed, the respiratory exchange ratio (RER), a key indicator of substrate utilization, was consistent with an enhanced contribution of carbohydrates to energy expenditure specifically at the end of the day, at a time at which wild-type animals were preferentially making use of lipid oxidation (Fig. 2h and Supplementary Fig. 2).

Analysis of the expression level of key ubiquitin ligases as well as Akt phosphorylation at Ser[473] in the skeletal muscle of mutant and control mice strongly suggested that the decrease of the lean mass observed in mutant mice did not involve either catabolic pathways or change in the insulin signaling in the skeletal muscle (Supplementary Fig. 1i, j).

These results indicate that the liver-specific deletion of *Lkb1* results in whole-body changes in body composition, with a lower lean mass, resulting in sarcopenia in the long term. The positive energy balance and preferential use of carbohydrate oxidation observed in the mutant animals may have contributed to their higher fat mass.

**Lkb1 controls hepatic gluconeogenesis and amino acid catabolism.** Studies based on inhibition of the gluconeogenic program have characterized Lkb1 as a suppressor of hepatic gluconeogenesis[7–10]. Consistent with these findings, pyruvate tolerance tests revealed higher levels of gluconeogenesis in Lkb1KO^livad mice, associated with higher levels of expression for several of the target genes of Foxo-1, a master transcriptional activator of hepatic gluconeogenesis[14,15] (Fig. 3a, b).

Given the complex metabolic phenotype of the mutant animals, which presented hyperglycemia, but with a defect in lean mass leading to sarcopenia in the long term, we performed proteomic analyses to decipher the role of Lkb1 in the control of liver metabolism. We used a label-free quantification (LFQ) method to characterize the liver proteome of Lkb1KO^livad mice and controls. This LFQ analysis was performed on the livers of fasted and refed animals. We identified 177 (63 upregulated and 114 downregulated) and 322 (165 upregulated, 156 downregulated) proteins differentially expressed (DE) between the

liver of mutant and control mice in the fasted and refed states, respectively ($-1<$FC (Log2) $>1$, adjusted $p$ value $<0.05$), (Supplementary Data 1 and Supplementary Data 2).

Ingenuity pathway analysis (IPA) identified gluconeogenesis and amino acid metabolism, and, more specifically, amino acid catabolism, as the top canonical pathways enriched in the DE proteins upregulated in the livers of fasted *Lkb1* mutant mice, and

in refed mutant mice (Fig. 3c, d). Most amino acids can be used as precursors of gluconeogenesis in the liver, as shown in Fig. 3e. In this scheme, we indicate the various enzymes involved in amino acid metabolism upregulated in the livers of the mutant animals. The heatmap of the DE proteins linked to amino acid catabolism (Tat, Pah, Hgd, Hpd, Kynu, Sds, Sdsl, Cth, Cbs, Pccb, Gnmt, Mat1a, Ahcy, Aass), aminotransferases (Got1, Agxt, Gpt2), amino

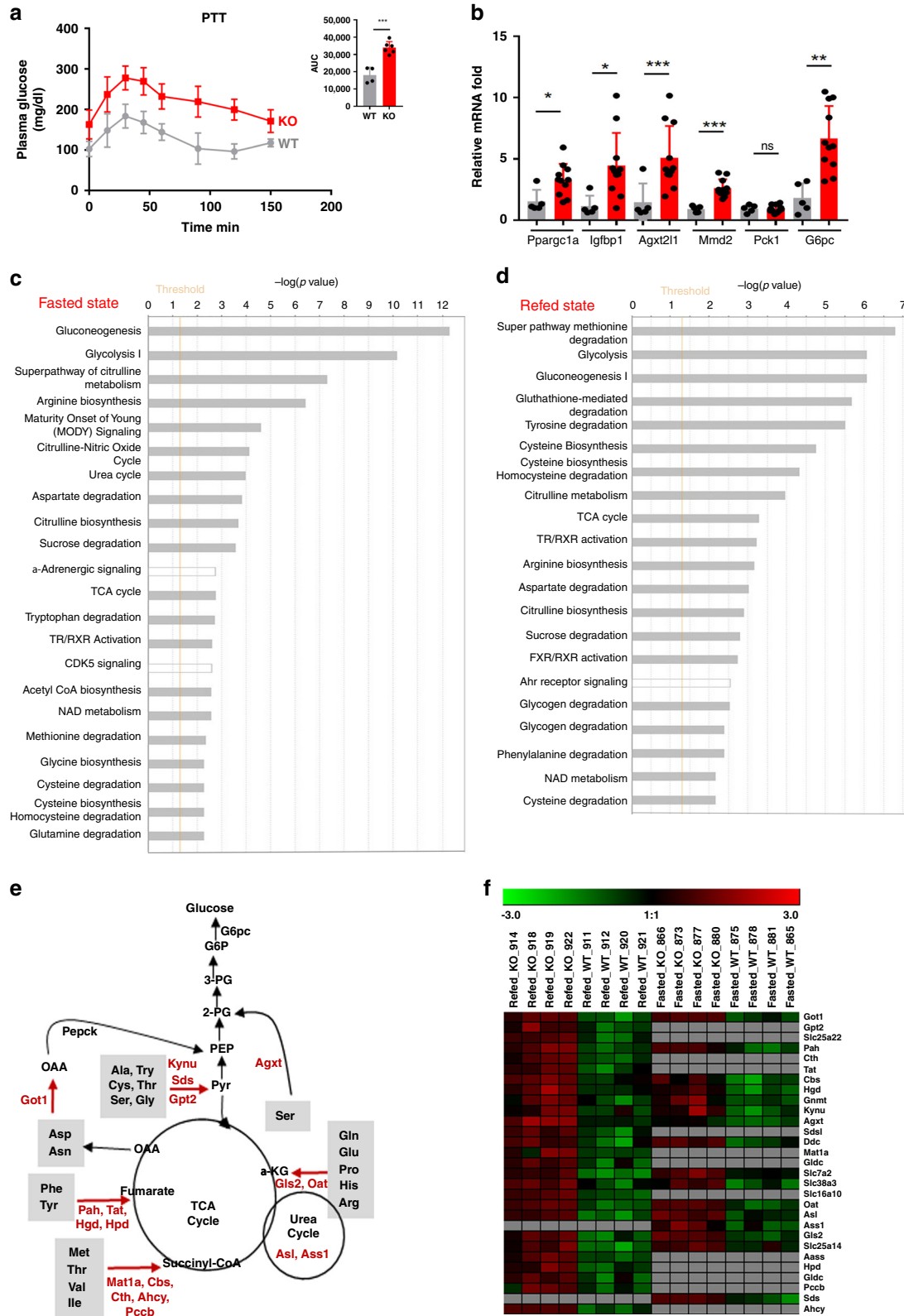

**Fig. 3 Lkb1KO^livad (KO) mice have high levels of liver gluconeogenesis and enriched hepatic amino acid catabolism. a** Pyruvate tolerance test (PTT) (KO: $n = 4$, WT: $n = 6$). The area under the curve (AUC) of blood glucose level is shown. Data are presented as mean values ± SD. $P$ values were determined by unpaired two-tailed $t$-test. ***$p \leq 0.001$. **b** RT-qPCR analysis of the expression of Foxo-1 target genes in fasted mice (KO: $n = 10$, WT: $n = 6$). Data are presented as mean values ± SD. $P$ values were determined by unpaired two-tailed $t$-test. ns: not significant; *$p \leq 0.05$; **$p \leq 0.01$; ***$p \leq 0.001$. **c, d** IPA analysis of the liver proteome of Lkb1KO^livad mice, showing the top-ranking enriched canonical pathways. The LFQ liver proteome analysis was done 15 days after tamoxifen injection on Lkb1KO^livad and control mice. The $p$ value for pathway enrichment was obtained in right-tailed Fisher's exact tests. The threshold for $-\log_{10}(p \text{ value})$ determines the probability of each biofunction being assigned to the dataset and the canonical pathway not being due to chance alone. **e** Scheme of the amino acid catabolic pathways for gluconeogenesis and the urea cycle. The proteins for which enrichment was noted in the livers of mutant mice are shown in red. **f** Heatmap showing the enrichment in proteins involved in amino acid catabolism, amino acid uptake and the urea cycle in both the fasted and refed states. This heatmap was generated with Genesis clustering software. Proteins not upregulated are shown in gray. Source data are provided as a Source data file.

acid uptake (Slc7a2, Slc25a15, Slc25a22, Slc38a3, Slc38a4) and the urea cycle (Oat, Gls2, Asl, Ass1) revealed a number of proteins deregulated in both the fasting and refed states, indicating that Lkb1 downregulates the levels of proteins involved in amino acid catabolism regardless of feeding status (Fig. 3f and Supplementary Data 1, Supplementary Data 2). Western-blot analyses on primary hepatocytes from Lkb1KO^livad and control mice revealed an upregulation of proteins involved in amino acid catabolism similar to that observed with mouse liver tissues (Supplementary Fig. 3a). This finding indicates that Lkb1 controls amino acid catabolism in a cell-autonomous manner.

Furthermore, a heatmap of the critical enzymes of gluconeogenesis (Pc, Pck1, Fbp1, G6pc) showed a strong upregulation of their protein levels in the refed state in mutant animals, highlighting the key suppressor role of Lkb1 in controlling hepatic gluconeogenesis not only during fasting periods, but also during the normal postprandial process (Supplementary Fig. 3b).

These results highlight a new role for Lkb1 as a suppressor of hepatic amino acid catabolism, in addition to its known role as a suppressor of gluconeogenesis, not only during the fasting period but also during the processing of nutrients after food intake.

**Lkb1 controls amino acid availability in hepatic gluconeogenesis.** The higher levels of hepatic amino acid catabolism in Lkb1KO^livad mice led us to explore the capacity of the liver to produce glucose from amino acids in *Lkb1*-deficient mice.

The main fate of alanine in the liver is to support gluconeogenesis. We, therefore, performed alanine tolerance tests (ATT) and determined glucose levels in the bloodstream after the IP injection of a bolus of alanine in fasted mice. In WT animals, alanine injection had a significant but modest effect on plasma glucose concentration, which rapidly returned to normal levels. However, in Lkb1KO^livad mice, we observed a large increase in glycemia, which remained high for more than 3 h after alanine injection (Fig. 4a).

The results obtained for primary cultures of hepatocytes from Lkb1KO^livad mice provided further evidence of the greater ability of *Lkb1*-deficient hepatocytes than of wild-type hepatocytes to produce glucose from amino acids. Hepatocytes from mutant mice produced larger amounts of glucose following the addition of pyruvate and lactate to the culture medium, consistent with the known role of Lkb1 as a suppressor of hepatic gluconeogenesis. However, glucose production levels were also higher in *Lkb1*-deficient hepatocytes following the addition of the three gluconeogenic amino acids tested: alanine, glutamine, and serine (Fig. 4b).

Consistent with these findings, measurements of metabolic fluxes in liver explants from fasted Lkb1KO^livad mice and controls incubated with [^14C]alanine showed that the hepatocytes of mutant mice performed both de novo glucose synthesis and glycogen synthesis from alanine more efficiently than those of wild-type mice (Fig. 4c).

This greater utilization of amino acids for de novo glucose synthesis in Lkb1KO^livad mice was associated with a lower plasma concentration of most amino acids and, for some, a lower hepatic content compared to controls (Fig. 5a, b). Surprisingly, alanine content was maintained in both the serum and liver, possibly due to the massive release of alanine by the muscles during fasting, at much higher levels than for any of the other amino acids[16,17]. Given the role of the muscle as the most important source of amino acids for gluconeogenesis during fasting and the lower lean mass of mutant mice observed on MRI, we also determined the amino acid content of the tibialis: in mutant mice, this muscle had an amino acid profile with deficits very similar to those observed in the liver (Fig. 5c). This lower muscle amino acid content is probably a direct consequence of the lower plasma amino acid concentration in these mice, as this concentration is known to affect protein turnover in muscle[18,19]. Hepatic and plasma urea and ammonia levels were similar in Lkb1KO^livad mice and controls, suggesting that the capacity of mutant animals to clear the nitrogen released by amino acid catabolism was preserved (Fig. 5d).

Collectively, our data demonstrate an enhancement of the ability of the liver of *Lkb1*-deficient mice to synthesize glucose from amino acids during fasting, but also during feeding, with the storage of this glucose in the form of glycogen. Our data also provide strong evidence that the control, by Lkb1, of the use of amino acids for gluconeogenesis in the liver controls the partitioning of amino acids between the liver and muscle. Hepatic Lkb1 thus appears to be a major player in the regulation of whole-body amino acid metabolism through its control of the use of amino acids in hepatic gluconeogenesis.

**Aminotransferases including Agxt are effectors of gluconeogenesis mediated by Lkb1.** Aminotransferases are frequently involved in the first step of amino acid metabolism for glucose production. The aspartate aminotransferase Got1 was one of the top-ranking upregulated DE proteins for all the nutritional states tested, and Gpt2 was also upregulated in the liver of refed mutant mice (Fig. 3f and Supplementary Data 1, Supplementary Data 2), identifying aminotransferases as potential critical effectors of the function of Lkb1 as a suppressor of amino acid-driven gluconeogenesis. We thus assessed the effect of amino-oxyacetic acid (AOA), a pan-aminotransferase inhibitor[20] on glucose homeostasis, in mutant and control animals in vivo. AOA treatment rescued the fasting hyperglycemia phenotype of mutant mice (Fig. 6a). Consistent with these data, AOA treatment led to a significantly lower peak blood glucose concentration in ATTs, and a faster return to basal levels than in untreated mutant mice (Fig. 6b).

We also investigated the involvement of alanine glyoxylate aminotransferase (Agxt) in the phenotype of Lkb1KO^livad mice. Indeed, *Agxt* was one of the four genes we identified as commonly induced in the liver of mice bearing the specific *Lkb1* deletion, not

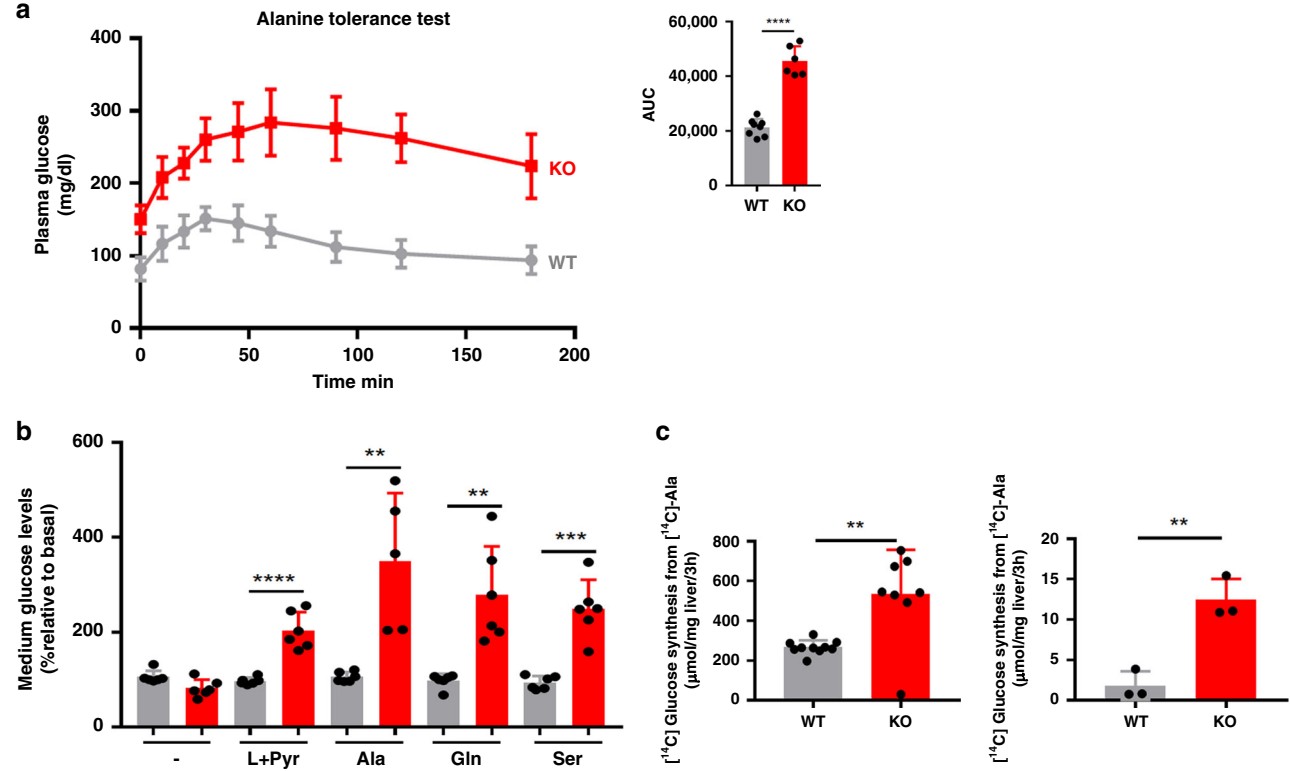

**Fig. 4 Liver gluconeogenesis from amino acids in Lkb1KO^livad (KO) mice. a** Blood glucose levels in alanine tolerance test (KO, $n = 6$, WT: $n = 8$). The area under the curve (AUC) of glucose level is shown. Mice were fasted for 24 h before the ATT. Data are presented as mean values ± SD. $P$ values were determined by unpaired two-tailed $t$-test. ***$p \leq 0.001$. **b** Glucose production from glutamine, alanine or serine in primary cultures of hepatocytes. Glucose production was normalized against protein content and is expressed as a percentage of the glucose produced by WT hepatocytes incubated in the absence of gluconeogenic substrates. Data are means ± SD of six biologically independent cell culture. $P$ values were determined by unpaired two-tailed $t$-test. **$p \leq 0.01$; ***$p \leq 0.001$. **c** [$^{14}$C]-glucose and [$^{14}$C]-glycogen synthesis from [$^{14}$C]-alanine in cultured liver explants from fasted mutant (KO, $n = 3$) and fasted control (WT, $n = 3$) mice. Data are means ± SD. $P$ values were determined by unpaired two-tailed $t$-test. **$p \leq 0.01$. For all the experiments described in this figure, animals were analyzed 15 days after the injection of tamoxifen. Source data are provided as a Source data file.

only in the adult liver (Lkb1KO^livad), but also in the embryonic liver[21] (Supplementary Data 3, Supplementary Fig. 4a). Agxt is also known to be involved in the synthesis of glucose from serine, in addition to its role in glyoxylate detoxification[22], as schematized on Fig.6c.

Western blotting and RT-qPCR analyses confirmed that Agxt was induced in the liver of mutant mice, in both the fasted and refed states, at both the protein and mRNA levels (Fig. 6d, e).

We investigated Agxt function, by studying Lkb1- and Agxt-deficient DKO mice obtained by crossing AgxtKO mice bearing a complete inactivation of Agxt[23] with Lkb1KO^livad mice (Supplementary Fig. 4b). DKO mice had a better fasting hyperglycemia phenotype than Lkb1KO^livad mice, but no improvement in postprandial hyperglycemia was observed highlighting that the function of Lkb1 may differ between the fasted and refed state (Fig. 6f). Metabolic analyses of the DKO mice revealed no major changes compared to the Lkb1KO^livad mice, specifically for the glycogen content, indicating that the sole Agxt deletion is not sufficient to rescue the glycogen phenotype of Lkb1KO^livad mice (Supplementary Fig. 4c–e, g–i). Follow-up of the mice for a longer period revealed much higher survival rates for DKO mice than for Lkb1KO^livad mice, with the lethal cachexic phenotype developing significantly later in DKO mice (Fig. 6f).

Thus, both the pharmacological inhibition of aminotransferases and genetic Agxt inactivation partly rescued the fasting hyperglycemia phenotype of Lkb1KO^livad mice. The deletion of Agxt also delayed the premature death of Lkb1KO^livad mice. Overall, these results indicate that aminotransferases including Agxt are key effectors of the suppressor function of Lkb1 in amino acid-driven gluconeogenesis.

**Lkb1 controls amino acid-driven gluconeogenesis via phosphorylation of RNA binding proteins**. We searched how Lkb1 may repress the hepatic amino acid-driven gluconeogenesis. We first looked at the possible involvement of Ampk. Using hepatocyte-specific Ampkα1/α2 null mice[24], we showed that Ampk did not control the expression of several enzymes of the amino acid metabolism such as Agxt, Got1 and Oat, strongly suggesting that Lkb1 controlled the hepatic amino acid-driven gluconeogenesis independently of Ampk (Fig.7a). To elucidate the mechanism by which Lkb1 suppress hepatic gluconeogenesis from amino acids, we used a quantitative phosphoproteomic approach to characterize the phosphoproteome of the liver of mutant Lkb1KO^livad in both the fasted and refed state. Our comprehensive analysis identified 7851 phosphopeptides from 1952 phosphoproteins (4383 phosphopeptides for 1387 proteins in fasted liver and 4690 phosphopeptides for 1414 proteins in refed liver). Differential expression analysis using Perseus software identified significant changes ($t$-test, $p$ value <0.01) of differentially expressed (DE) phosphoproteins in the fasted and refed state. We also retained all the phosphopeptides that were not quantified (designated NaN in Supplementary Data 4) in any of the KO samples whereas quantified in the WT samples, and vice versa (WT versus KO). They likely represent phosphopeptides that were dramatically controlled by Lkb1 (Supplementary

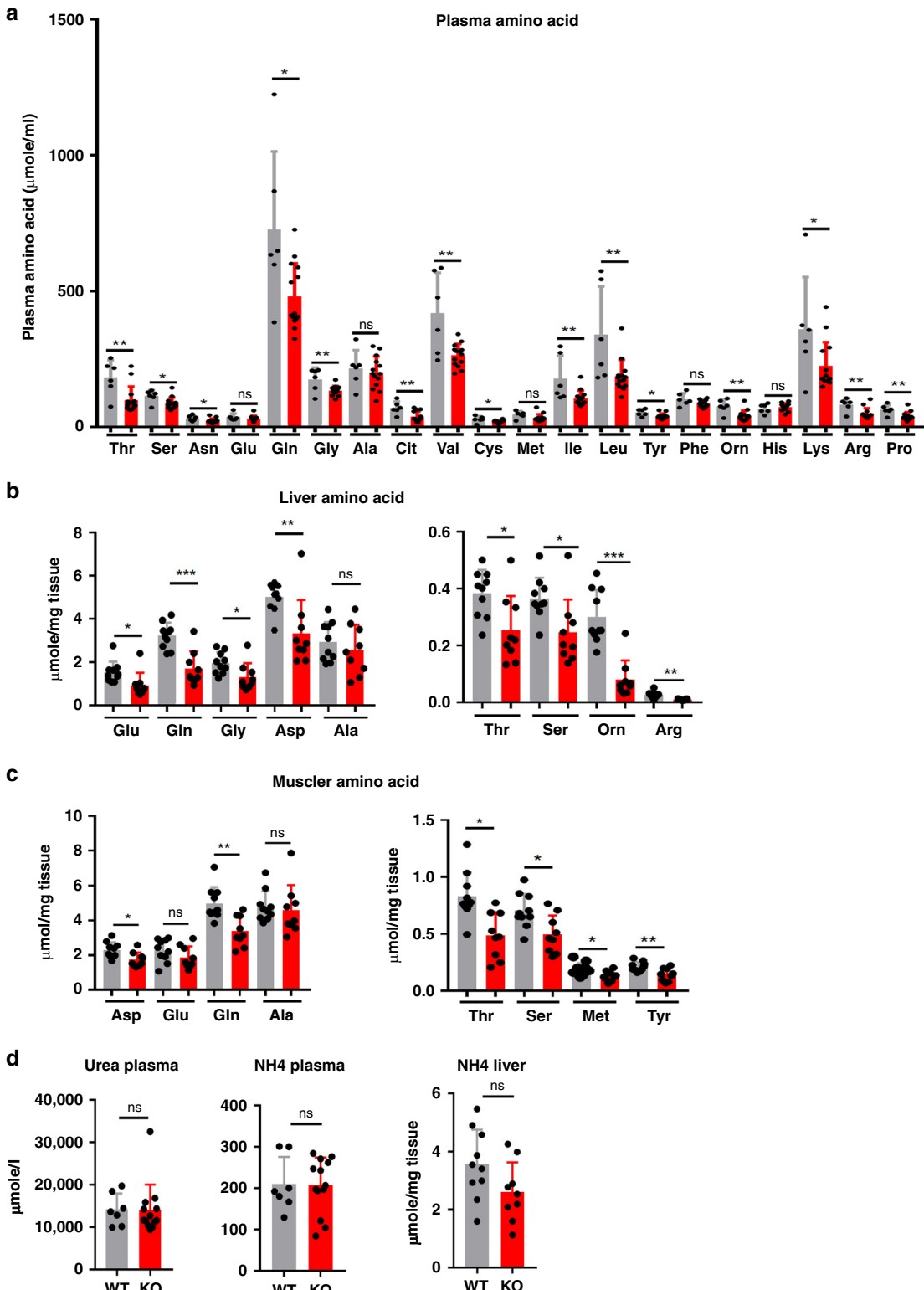

**Fig. 5 Plasma, liver, and muscle amino acid content of Lkb1KO^livad mice. a** Plasma amino acid concentration in fasted mutant (KO: $n = 13$, red bar) and fasted control (WT: $n = 7$, gray bar) mice. **b** Hepatic amino acid content of fasted mutant mice (KO: $n = 10$, red bar) and fasted control mice (WT: $n = 9$, gray bar). **c** Muscle (tibialis) amino acid content of fasted mutant mice (KO: $n = 10$, red bar) and fasted control mice (WT: $n = 9$, gray bar). **d** Ammonia and urea level in the plasma and ammonia level in the liver of Lkb1KO^livad(KO, $n = 13$, red bar) and controls (WT, $n = 7$, gray bar). All data are presented as mean values ± SD. $P$ values were determined by unpaired two-tailed $t$-test. ns: not significant; *$p \leq 0.05$; **$p \leq 0.01$; ***$p \leq 0.01$. For all the experiments described in this figure, animals were analyzed 15 days after the injection of tamoxifen.

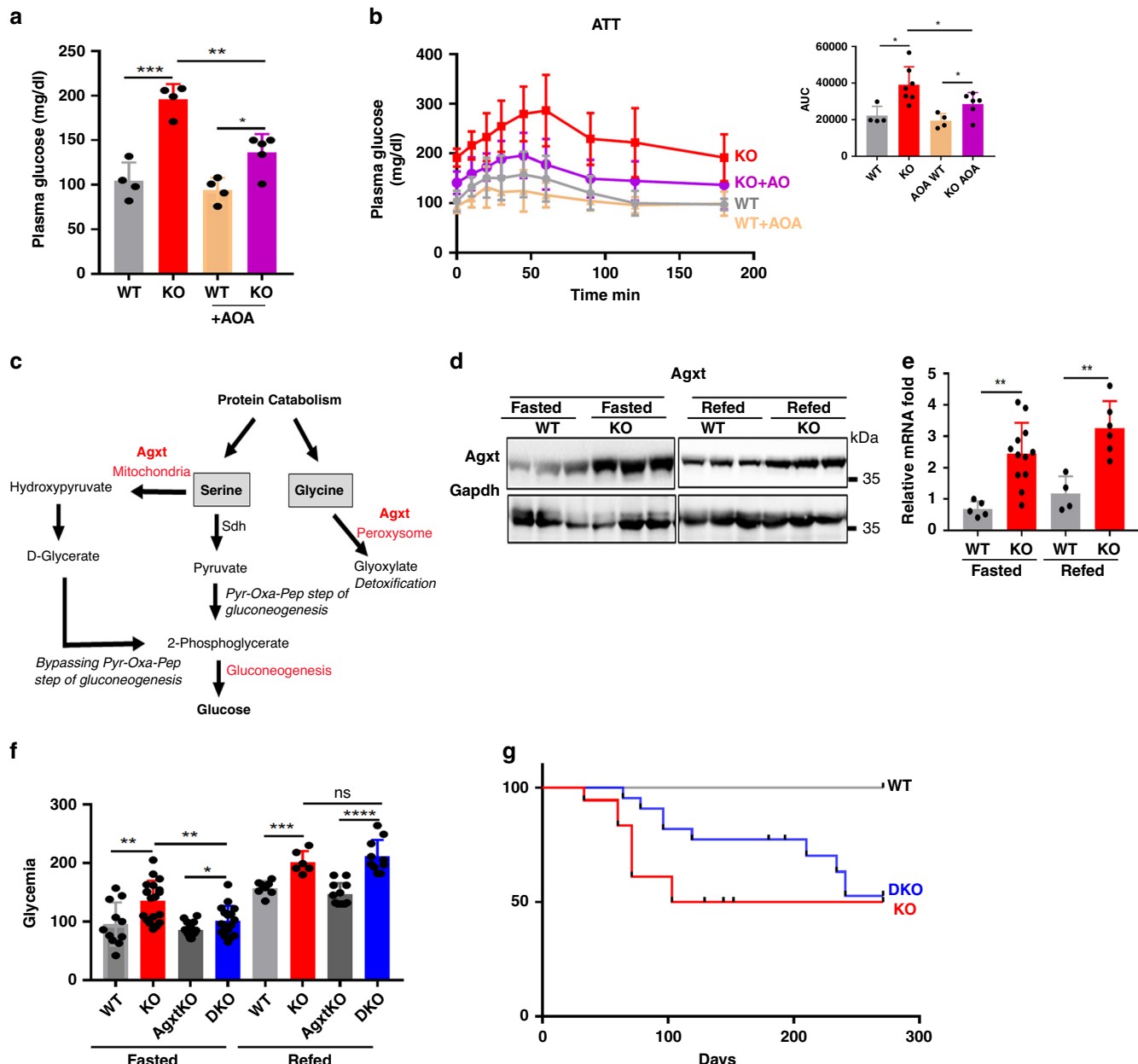

**Fig. 6 Aminotransferases including Agxt are crucial effectors of the suppressor function of Lkb1 in amino acid-driven gluconeogenesis. a**, **b** Pharmacological inhibition of aminotransferase, in vivo, by AOA. Lkb1KO[livad] (KO) and WT mice were left untreated (−) or were treated with AOA (10 mg/kg) 2 h before the determination of fasting blood glucose levels (**a**) or the alanine tolerance test (**b**). The area under the curve (AUC) of glucose level in the ATT is shown. AOA-treated animals, KO: $n = 5$, WT: $n = 6$. Animals not treated with AOA, KO: $n = 5$, WT: $n = 4$. Mice were fasted for 22 h before treatment with AOA or were left untreated for the same time period. **c** Scheme of the dual function of Agxt. **d**, **e** Agxt protein and mRNA levels in Lkb1KO[livad] (KO) mice. **d** immunoblot of Agxt in fasted and refed mutant (KO, $n = 3$) and control (WT, $n = 3$) mice. Representative blot of two independent experiments. **e** Agxt gene expression levels, as assessed by RT-qPCR in fasted (KO: $n = 10$, WT: $n = 6$) and refed (KO: $n = 7$, WT: $n = 5$) mice. **f**, **g** Phenotypes of mice lacking both Lkb1 and Agxt in the hepatocytes (DKO), as compared with mice with a single deficiency of Lkb1 in hepatocytes (Lkb1KO[livad], KO) and controls (WT and AgxtKO). **e** Glycemia in fasting (WT: $n = 11$, KO: $n = 19$, DKO: $n = 17$, AgxtKO: $n = 18$) and refed (WT: $n = 8$, KO: $n = 6$, DKO: $n = 10$, AgxtKO: $n = 12$) mice. **f** Kaplan–Meier survival curves for Lkb1KO[livad] mice ($n = 18$), mice lacking both Lkb1 and Agxt (DKO $n = 23$) and controls (WT, $n = 10$). All graphical data are means ± SD. P values were determined by unpaired two-tailed t-test. ns: not significant. $*p \leq 0.05$; $**p \leq 0.01$; $***p \leq 0.001$; $****p \leq 0.0001$. Source data are provided as a Source data file.

Data 4). Altogether we found 739 DE phosphosites (639 down-regulated, 100 upregulated) in the fasted state and 477 DE phosphosites (59 downregulated, 418 upregulated) in the refed state. Among the common molecules 78.5% have an opposite regulation. On a protein scale, this phosphosites matched with 415 DE proteins for the fasted state, 308 for the refed state with 121 common proteins. (Supplementary Data 4). As expected, in fasted mutants, we observed that most of the DE proteins had reduced phosphorylation, Lkb1 being a kinase. However, in the refed mutants, most of the DE proteins were hyperpho-sphorylated, highlighting that Lkb1 acts differently between the fasted and refed states (Supplementary Data 4). Importantly, less than 5% of the significantly regulated phosphopeptides were located on proteins whose expression levels were observed to

change confirming that the majority of the observed changes in the phosphoproteome are not driven by changes in protein expression (Supplementary Fig. 5a). IPA analysis revealed an enrichment for proteins involved in the regulation of RNA post-transcriptional modification and protein synthesis (Fig. 7b), indicating that Lkb1 could control amino acid catabolism at a post-transcriptional level. Consistent with this result, we did not

find any increase in the expression of genes encoding proteins involved in hepatic amino acid metabolism in a microarray analysis on the livers of fasted LKBKO[livad] mice, except for *Agxt* (Supplementary Data 3), indeed suggesting a control of protein expression at the translational level. Because the control of mRNA translation relies around messenger ribonucleoprotein (mRNP) complexes, we focused on the RNA binding proteins

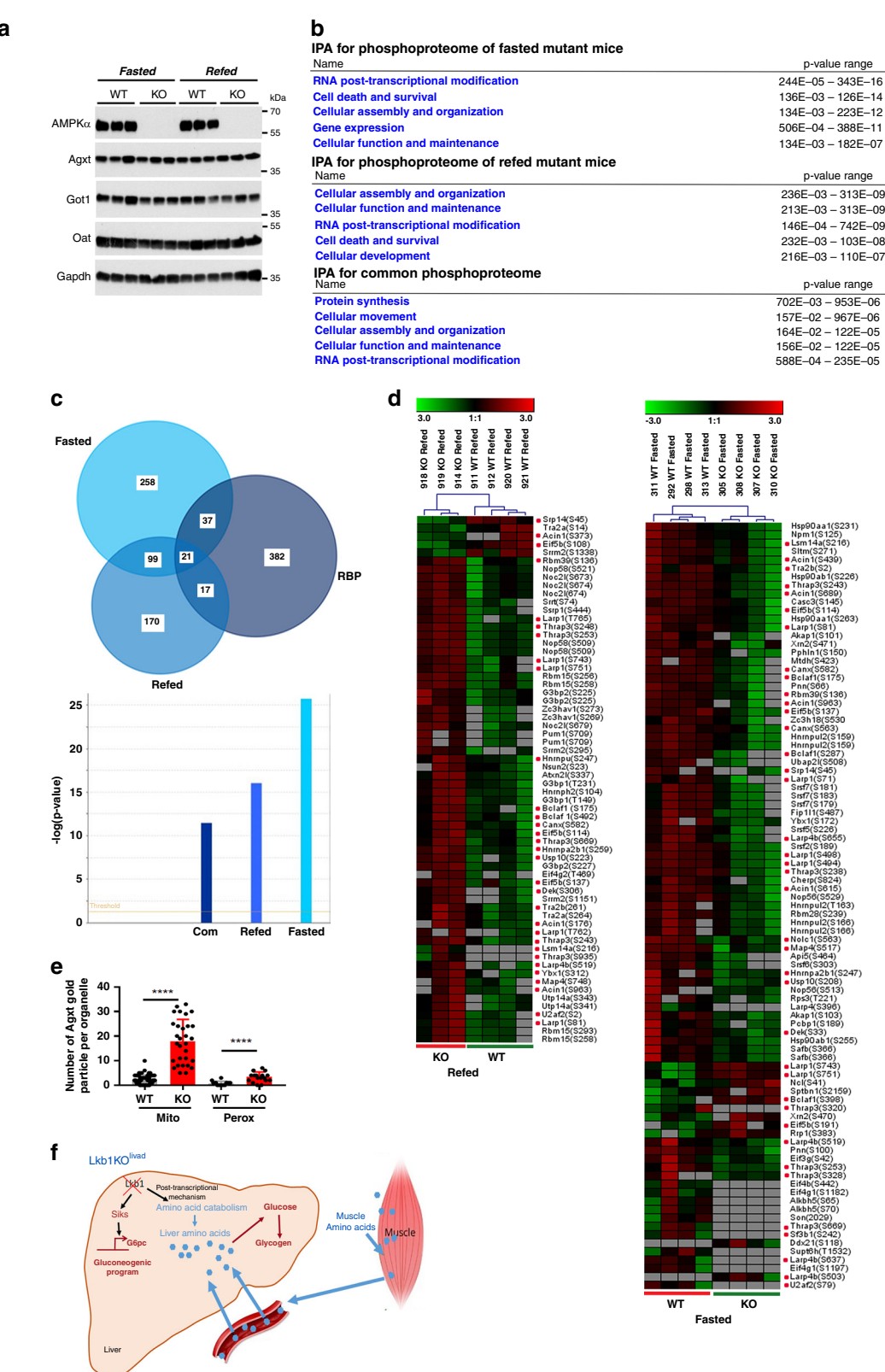

**Fig. 7 Lkb1 controls the hepatic amino acid-driven gluconeogenesis independently of Ampk and likely via the phosphorylation of RNA binding proteins.**
**a** Immunoblot of liver of mutant Ampk mice. Liver of double knockout of Ampkα1 and Ampkα2 catalytic subunits (KO, n = 3) and controls (WT, n = 3) were analyzed by western blot analysis for expression of Ampk, Agxt, Got1, and Gapdh. Representative blot of two independent experiments. **b** IPA analysis of the significantly deregulated phosphoproteins both in the fasted and refed state. A comprehensive phosphoproteomic analysis of the liver of Lkb1KO^livad mutant and control was performed and the significant differentially expressed phosphoproteins were analyzed for enrichment in GO terms. **c** Venn diagram comparing the canonical RBP repertoire common to HuH7, HEK293, and Hela cells[28] with the significantly DE phosphoproteins of liver of both fasted and refed mutant mice. Enrichment for RBP was done using IPA. **d** Heatmap of the significantly deregulated phospho-RBP in livers of Lkb1KO^livad and control animals. The phospho-RBP that were shared between the fasted and refed state were highlighted with a red point. Note that most of the phophosites differed and had opposite phosphorylation status between the two nutritional states for most of the phosphoproteins. Heatmap was done using the Genesis software. **e** Subcellular distribution of Agxt in liver of mutant Lkb1KO^livad and control mice. Subcellular distribution of immunoreactive Agxt was determined by post-embedding protein A gold immuno-electron microscopy. Labeling intensity was obtained by counting the number of gold particles in each organelles. A total of at least 50 fields were analyzed for both the mutant and control liver. Data are represented as means ± SD of two mice per group. P values were determined by unpaired two-tailed t-test. p ≤ 0.0001. **f** General scheme summarizing how Lkb1 controls liver gluconeogenesis and more specifically the amino acid driven gluconeogenesis. Source data are provided as a Source data file.

(RBP) that are key actors of the mRNP complexes[25]. Since mRNP are highly dynamic and phosphorylation is a key regulatory mechanisms for mRNP remodeling[26,27], the identification of the phophosites of the RBP that are controlled by Lkb1 appears to be an interesting clue to understand how Lkb1 control the hepatic amino acid catabolism at the translational level. We compared the DE phosphoproteins identified in the fasted and refed state to the canonical RBP repertoire, published by Beckmann et al[28] and found a strong enrichment in RBP in both the fasted and refed state (Fig. 7c). 22 RBP were common to the fasted and refed state although with different phosphosites and opposite regulation for 82% of them (Supplementary Data 4). Heatmap of the significant differentially phosphorylated RBP confirmed that most of the phospho-RBP controlled by Lkb1 were downregulated in the fasted liver while they were upregulated in the refed liver and involved different phosphopeptides for each common phospho-proteins (Fig. 7d, Supplementary Fig. 5b). Interestingly, we found in the phosphoprotein network of Lkb1 many regulators of translation including eukaryotic initiation and elongation factors (Eif3b, Eif3g1, Eif4b, Eif4g1, Eif4g2, Eif5b, Eef1d) as well as factors such as Larp1, Larp4, Lsm14, Npm1, Nolc1, Pum1, Ybx1 (Fig. 7d and Supplementary Data 4). As a further step towards identifying a regulatory link between Lkb1-dependent amino acid catabolism and RBP biology, we constructed a network map integrating the interactions between: (1) Lkb1, (2) the differentially phosphorylated RBP and (3) the differentially phosphorylated kinases and phosphatases, (4) the proteins of amino acid catabolism and involved in gluconeogenesis that are controlled at the translational level. This approach identified three main clusters of interactions, i.e Lkb1 with the phosphorylated kinases/phosphatases, the phosphorylated RBP and the Lkb1 metabolic targets (Supplementary Fig. 5c). This analysis revealed distinct mechanisms by which Lkb1 controls expression of its metabolic targets in the fasted and refed state. While direct interactions between Lkb1 and phosphorylated RBP, such as Hsp90aa1 as well as indirect paths involving kinases could be observed in the fasted state, only indirect paths could be observed in the refed state between Lkb1 and the phosphorylated RBP (Supplementary Fig. 5c).

Agxt performs different functions of glyoxylate detoxification and neoglucogenesis depending on its location in peroxisomes and mitochondria. In the mitochondria, it is mainly involved in neoglucogenesis[29]. The variable localisation of Agxt depends of alternative transcription or alternative initiation sites that allows expression of N-terminal mitochondrial and C-terminal peroxisomal targeting sequences[29]. Interestingly, in rats, when fed on high protein diets or given gluconeogenic stimuli, such as glucagon, there is a large increase in the synthesis of mitochondrial, but not peroxisomal Agxt[30]. We thus performed

colloidal gold immuno-electron microscopy to study the cellular distribution of Agxt in the liver of mutant Lkb1KO^livad and controls. Expression of Agxt was dramatically increased in both mitochondria and peroxisomes of mutant animals compared to controls (Fig. 7e and Supplementary Fig. 5d) indicating that Lkb1 did not control the specific loaclization of Agxt but rather its whole protein expression level.

Overall, our data showed that Lkb1 controlled amino acid-dependent gluconeogenesis independently of Ampk but via a phosphoprotein network of RBPs and highlighted the involvement of different mechanisms during the fasting or the postprandial state.

## Discussion
Hepatic Lkb1 was already known to be a key gluconeogenic suppressor acting by downregulation of the gluconeogenic gene program[7,8,10]. Here, we demonstrate that Lkb1 acts as a suppressor of amino acid driven gluconeogenesis in the liver, in both the fasting and postprandial states. Mice with a hepatocyte-specific deletion of Lkb1 had higher levels of hepatic amino acid catabolism, to provide the carbon skeleton for glucose synthesis. The deregulation of hepatic amino acid extraction for gluconeogenesis was associated with a lower plasma amino acid concentration, with effects on skeletal muscles, which serve as the principal source of amino acids for gluconeogenesis in the fasting state. These findings are consistent with the lower lean body mass of the mutant animals, a phenomenon that was so intense that it led to sarcopenia, cachexia, and death in the long term. Inactivation of hepatic targets of Lkb1—aminotransferases and Agxt—partly rescued the fasting hyperglycemia phenotype, identifying these targets as effectors of the suppressor function of Lkb1 in amino acid-driven gluconeogenesis. Finally, we showed that Lkb1 controls the hepatic amino acid-driven gluconeogenesis independently of Ampk and suggested that Lkb1 regulates amino acid catabolism at the translation level through the phosphorylation of RNA binding proteins. A scheme of the mechanisms by which Lkb1 controls amino acid-driven gluconeogenesis and the glycogen content in the liver is shown in Fig. 7f.

The liver is a major site of amino acid disposal. It is the only organ with the enzymatic equipment required for the metabolism of all amino acids and with a complete urea cycle for efficient handling of the potentially toxic ammonia generated by amino acid breakdown. The regulation of amino acid metabolism in the liver is crucial, because the high plasma amino acid concentration occurring the postprandial period may be toxic and the body has no storage mechanism for amino acids like those for fatty acids, as triglycerides, or for glucose, as glycogen. Excess amino acids are converted into glucose in the liver, and gluconeogenesis from

amino acids is a normal postprandial process not limited to fasting periods[3,31].

One unforeseen consequence of unrestrained hepatic gluconeogenesis from amino acids was the impact on muscle physiology observed in mice with a hepatocyte-specific *Lkb1* deficiency. Our data indicated that the decrease in lean body mass was neither associated with activation of catabolic pathways nor changes in insulin signaling in skeletal muscle. This led us to hypothesize that the decrease in lean body mass observed in the mutant animals is an indirect consequence of the higher levels of amino acid extraction for glucose synthesis in these animals, leading to a decrease in plasma amino acid concentration. Indeed, plasma amino acid availability has been shown to make a major contribution to muscle protein turnover. High amino acid availability, as observed in the postprandial state, favors muscle anabolism by activating protein synthesis through control over the translation initiation step of protein synthesis[32], whereas low amino acid availability leads to an inhibition of muscle protein synthesis, resulting in a net release of amino acids from the muscle[33]. Indeed, mice with a knockout of the *Klf15* transcription factor, an important regulator of branched-chain amino acid catabolism[34], have low levels of hepatic amino acid catabolism associated with high hepatic and plasma amino acid concentrations, resulting in a significantly higher lean mass than observed in control animals[35]. This mutant is, thus, the mirror image of Lkb1KO[livad] mice. Similarly, glucagon is known to stimulate hepatic amino acid catabolism for gluconeogenesis[36]; in all models of glucagon deficiency, the phenotype of the mutant mice mirrors that of Lkb1KO[livad], with the operation of a similar amino acid catabolism program and increases in the levels of most amino acids in the plasma, and many amino acids in the liver[37–39]. However, the impact on muscle is less striking in these mutants, which display only a slight increase in lean mass[40]. However, we cannot exclude the possibility that mechanisms other than amino acid availability per se are involved in these modifications to relationships between the liver and muscle observed in Lkb1KO[livad] mice; indeed, hepatic Lkb1 may control the synthesis of hepatokines acting on peripheral tissues.

The accumulation of glycogen in fasted Lkb1KO[livad] mice was surprising and revealed a defect in glycogen use during fasting in these mice. Fasted mutant mice displayed much lower levels of phosphorylation of the glycogen phosphorylase (Pygl), indicative of lower levels of glycogen degradation. The reason for this inactivity of Pygl remains unknown, but hyperglycemia has been shown to be a strong suppressor of glycogenolysis[41]. Of note, we observed an increase in the glycogen synthase (Gys2) content both in the fasted and fed state in the liver of mutant animals (Fig. 1f and Supplementary Data 1, Supplementary Data 2). Our data indicate that the activity of Gys2 may not be controlled by a phosphorylation event, but only by an increase in Gys2 total protein level. In the fasted state, Gys2 was not identified as differentially phosphorylated between mutant and control in our quantitative phosphorproteomic analysis. In the refed state, we found an increase in Gys2 phosphorylation at the Ser11 site in mutant liver, but this phosphosite is described as inhibitory for Gys2 activity (Supplementary Table 4). Flux analyses with [14]C-labeled alanine demonstrated higher levels of glycogen synthesis in fasted *Lkb1*-deficient hepatocytes, revealing the ability of these cells to synthesize glycogen from alanine via the indirect pathway[2]. Thus, both a decrease in glycogenolysis and an increase in glycogen synthesis explain the persistence of glycogen during fasting in these mutants. The enhanced rerouting of amino acids to liver glycogen observed in mutant animals probably explains the moderate increase in hyperglycemia observed in these animals despite unrestrained amino acid-driven gluconeogenesis.

Our data showed that Lkb1 controls hepatic amino acid-driven gluconeogenesis both in the fasted and refed state. These results highlighted a critical role of Lkb1 in the control of postprandial gluconeogenesis. Although postprandial gluconeogenesis is critical for the body to cope with excess of dietary amino acids, this aspect and the contribution of Lkb1 in its control have been poorly studied. We have shown that it is independent of the Ampk signaling. The in silico analysis of our phosphoproteomics data sets identified a Lkb1-dependent network of phosphorylated RBP that supports the control of the amino acid catabolism at the translation level. Interestingly, the mechanism by which Lkb1 controls liver gluconeogenesis from amino acids during the fasting state differ from that used during postprandial gluconeogenesis. Even if similar phosphoproteins are used in the two nutritional states, the phosphorylation sites and the regulation were clearly distinct.

We identified aminotransferases as effectors of the function of Lkb1 in the control of amino acid-driven gluconeogenesis. The fasting hyperglycemia phenotype of the mutant mice was partially rescued by treatment with a pan inhibitor of aminotransferases, AOA, consistent with the critical role of aminotransferases, which are frequently involved in the first step of amino acid catabolism. Consistent with our data, a recent study showed that inhibition of the hepatic alanine transaminase reduced amino acid gluconeogenesis and was associated with a reduced postprandial blood glucose and the rescue of the hyperglycemia phenotype of different models of diabetes[42]. In addition our data highlight a role for Agxt a serine-pyruvate transaminase involved in both gluconeogenesis from serine[22] and glyoxylate detoxification. *AGXT* mutations cause primary hyperoxaluria[43], however, no changes of glucose metabolism have been reported in patients with such mutations. In our experiments, *Agxt* deletion partially rescued the fasting hyperglycemia phenotype of Lkb1KO[livad] mice. We were able to monitor the phenotype of the DKO mice, in which the *Agxt* gene was deleted, over longer time periods. *Agxt* deletion greatly delayed the development of cachexia in Lkb1KO[livad] mice. This result highlights the crucial role of gluconeogenesis deregulation in the development of the lethal phenotype in mutant mice. Interestingly, recent studies have identified transaminases as effectors of other metabolic functions of Lkb1 involved in the epigenetic reprogramming occurring during malignant transformation[44], or during the glial fate specification of neural crest cells[45]. Together with our data, these results highlight a key unsuspected role of aminotransferases in Lkb1 function.

Hepatic gluconeogenesis is frequently deregulated in DT2M, so the identification of Lkb1 as a suppressor of hepatic amino acid-driven gluconeogenesis should open up new avenues for the treatment of DT2M.

## Methods

**Ethical compliance statement**. All animal procedures were carried out according to French legal regulations (Ministère de la Recherche, de l'Enseignement Supérieur et de l'Innovation) and approved by ethics committee at the University Paris Descartes (Projet APAFIS 8722 and 8612).

**Mice**. Mice with an inducible specific deletion of *Lkb1* in hepatocytes (Lkb1KO[livad] mice) were obtained by crossing *Lkb1*[fl/fl] mice (FVB/N background)[46] with TTR-CreTam (FVB/N background) mice expressing an inducible Cre-recombinase under the control of the hepatocyte-specific transthyretin promoter[47]. The control mice for this study were their Cre-negative littermates (*Lkb1*[fl/fl]; Cre[−]). AgxtKO mice (C57Bl6/N background) were a gift from Anja Verhulst (University of Antwerp, Belgium) and Eduardo Salido (Tenerife, Spain)[23]. The DKO mice were obtained by crossing LKBKO[livad] mice with AgxtKO mice. Liver double knockout of *Ampkα1* and *Ampkα2* catalytic subunits was achieved by crossing *Ampkα1lox/lox* (C57Bl6/N background) mice with *Ampkα2lox/lox* (C57Bl6/N background) mice then *Alfp-Cre* transgenic mice (C57Bl6/N background) to generate *Ampkα1lox/lox,α2lox/lox* (control) and *Ampkα1lox/lox,α2lox/lox-Alfp-Cre* (liver *Ampkα1/α2* KO) mice[24].

Mice were housed in colony cages in SPF conditions, under a 12-h light/12-h dark cycle, in a controlled-temperature environment (21 °C) with 50% humidity. They were fed *ad libitum* with a standard laboratory chow diet (65% carbohydrate, 11% lipids, and 24% proteins; SAFE 03, FRANCE). In most cases, animals were studied either directly after 18 h of fasting (fasted state) or 4 h after the initiation of refeeding with a chow diet (refed state). For long-term experiments, animals were fed with the standard chow diet.

We injected two doses of tamoxifen (1.5 mg/mouse of tamoxifen Sigma-Aldrich, T5648) in corn oil IP into 8- to 12-week-old mice. Most studies were performed on male mice 15 days after injection. For the survival analysis, both males and females were monitored for up to 300 days. For all experiments, age-matched animals were used as controls. For protein, RNA and biochemistry studies, livers were removed after each experiment and frozen in liquid nitrogen. For information on the number of animals (*n*) in each experiment, please refer to the corresponding figure legend.

For AOA (O-(carboxymethyl)hydroxylamine hemichloride, Sigma-Aldrich C13408-1G) was dissolved in PBS, and injected into the animals via the IP route (10 mg/kg) 2 h before blood glucose determinations or alanine tolerance tests.

**Serum and liver biochemistry**. Serum samples were obtained by the immediate centrifugation of blood collected from the orbital sinus into heparin or EDTA. Plasma for biochemical or hormonal analyses was stored at −70 °C until use. Insulin and glucagon were determined with the MO DIABETES INSULIN SET and MO DIABETES GLUCAGON SET (BIO-RAD), respectively, by immunoassay on a Luminex apparatus (Biorad). The kits for determining alanine aminotransferase (ALAT) and triglyceride (TG) levels were purchased from DiaSys. Capillary blood glucose level was determined with an Accu-Check II Glucometer (Roche Diagnostic).

**Glycogen quantification**. The liver was homogenized and deproteinized by treatment with 4% (w/v) perchloric acid treatment. Hepatic glycogen content was then assessed as described by Roehrig and Allred[48]. Glycogen was first hydrolyzed with α-amyloglucosidase, and the glucose generated was then converted into 6-phosphogluconolactone by hexokinase treatment followed by glucose-6-phosphate dehydrogenase treatment in the presence of NADP. NADPH production was then assayed by spectrophotometry at 340 nm.

**Glucose, pyruvate, and alanine tolerance tests**. The IP glucose tolerance test (GTT) was performed on fasted male mice. Animals received an IP injection of 2 g of glucose/kg body weight (BW). Blood was drawn from the tail vein. Blood glucose levels were determined at various time points after glucose injection with the Accu-Check II Glucometer (Roche Diagnostic). Similar protocols were used for the pyruvate tolerance test (PTT) (injection of 2 g pyruvate/kg BW) and the alanine tolerance test (ATT) (injection of 2 g alanine/kg BW). The fasting period was 18 h for GTT and PTT, but 24 h for ATT.

**Analyses of energy expenditure and body composition**. Mice were analyzed for total energy expenditure (EE), oxygen consumption (VO₂) and carbon dioxide production (VCO₂), respiratory exchange rate (RER, VCO₂/VO₂), food intake (g), and spontaneous locomotor activity (beam breaks), in indirect calorimetry cages with bedding, food, and water (Labmaster; TSE Systems, Bad Homburg, Germany), as previously described[49]. The mice were individually housed before tamoxifen induction, and had free access to food and water *ad libitum*, with the lights on from 7 a.m. to 7 p.m. and an ambient temperature of 22 °C. All animals were allowed to acclimate for one week in calorimetry cages before experimental measurements.

Body mass composition (lean tissue mass, LBM), fat mass, free water and total water content were analyzed by MRI with the EchoMRI 100 system (Whole Body Composition Analyzers, EchoMRI, Houston, USA) according to the manufacturer's instructions[50].

Analysis was performed in Excel XP, on the extracted raw value of VO₂ (ml/h), VCO₂ (ml/h), and energy expenditure (kcal/h). Subsequently, each value was expressed by total body weight or total lean body mass measured by EchoMRI.

For the respirometry analysis of the DKO mice compared to KO mice, metabolic rates were measured using an 8-cage Promethion metabolic phenotyping system (SSI) as described in Lark et al.[51]. Animals had free access to drink and food hoppers in an ambient temperature of 22 ± 0.5 °C with light from 7 am to 7 pm. Mouse cage behavior, including roaming (XYZ beam breaks), food and water intake (to 1 mg) were monitored continuously at a sample rate of 1 sample/sec for all sensors and cages simultaneously via an error-correcting control area network (CAN). Air was pulled from the cages at a controlled mass flow rate of 2 L/min. O₂ and CO₂ were continuously monitored for assessment of EE. Air calibration were made accordingly to the manufacturer using a 100% nitrogen as zero reference and with a span gas containing of known concentration of CO2 (Air Liquide, S.A. France). Data were stored in raw, unprocessed form for later analysis using analysis scripts run on ExpeData analytical software (SSI). This allowed complete and traceable control of the analytical process, the equations used, the baselining algorithms employed, and all other aspects of data transformation and final data extraction.

**Immunoblot analysis**. Total protein extracts were obtained from 100 mg of frozen mouse liver or from mouse primary hepatocytes homogenized in lysis buffer (50 mMTris-HCl, pH 7.5, 150 mM NaCl, 5 mM EDTA, 30 mM Na₄P₂O₇, 50 mMNaF, 1% Triton, 1 mM DTT, protease inhibitor cocktail (Pierce #32953, Thermo Fisher Scientific)) supplemented with phosphatase inhibitor cocktail (Pierce #8867, Thermo Fisher Scientific) in a bead mill, with the Tissue Lyser disruption system (Qiagen, Hilder, Germany). Proteins were resolved by SDS–PAGE, transferred to nitrocellulose and blocked by incubation with 5% BSA or 5% milk. Blots were incubated with specific primary antibodies overnight at 4 °C, washed, incubated with the corresponding horseradish peroxidase-conjugated secondary antibodies (Cell Signaling) and developed by enhanced chemiluminescent techniques (Thermo Fisher Scientific). Primary antibodies were obtained from Cell Signaling Technologies (LKB1: clone D60C5, 1:1000; p-Ampk (T172), 50081, 1:1000; Ampk, 5831, 1:1000; Akt, p-Akt (S473), 4060, 1:1000; Gys2, 3886, 1:1000), Abcam (Oat, ab137679, 1:2000; Agxt, ab178708, 1:2000), and Santa Cruz (Gapdh, FL-335, 1:1000; Got1, sc-515641, 1 :1000), p-Pygl (S15), 1:100 was obtained from MRC PPU reagents and services. The antibody against Agxt for the immune gold analysis was from Abcam, ab178708, 1:200).

The bands on the immunoblots were quantified with FUJI multigauge software after incubation with a perioxidase-coupled secondary antibody and densitometric analysis.

**Histology and immunohistochemistry**. Mouse livers were cut into 3 mm-thick sections and fixed by incubation in 10% formalin for 12 h. They were then embedded in paraffin. For morphological analysis, 2 μm sections were cut, dewaxed, and stained with hemalun and eosin. Periodic acid-Schiff (PAS) staining was performed with 0.5% periodic acid solution, followed by Schiff's reagent and counterstaining with hematoxylin solution.

IHC staining for dystrophin was performed as described in Guerci et al[52].

**Agxt immunogold analysis**. After fixation (1% glutaraldehyde), tissues were washed 3 times with phosphate buffer, embedded in sucrose, and frozen in liquid nitrogen. Immunochemical reactions were performed on thin sections collected on grids according to the method of Bendayan[53]. After a brief incubation of the sections with 0.1% BSA and 15% normal goat serum, they were labeled with the Agxt polyclonal rabbit antibodies diluted in Tris-buffered saline (TBS) containing 1% BSA and 4% normal goat serum for 2 h at 22 °C, washed 3 times with TBS containing 0.1% BSA, and then incubated with goat antirabbit-gold (10 nm) for 1 h at 22 °C. The sections were counterstained with uranyl acetate.

Acquisitions were performed with a JEOL 1011 transmission electron microscope with an ORIUS 1000 CCD camera.

**RNA extraction and quantitative real-time PCR analysis**. Total RNA was extracted from mouse tissues and cell lines with Trizol Reagent (Life Technologies) according to the manufacturer's protocol. Reverse transcription was performed with 1 μg of total RNA and the Transcriptor First Strand cDNA Synthesis Kit (Roche Diagnostics), with random hexamer primers. Quantitative PCR was performed with the Light Cycler 480 Sybr Green I Master kit (Roche) and specific primers (Eurogentec) on a Light Cycler 480 thermocycler (Roche). RNA levels were calculated by the 2(-Delta Ct) method, with 18S as the internal control, relative to RNA levels in control littermates. The PCR primers used are: *Ppargc1a*, 5′-TGAAAGGGCCAAACAGAGAG-3′ (forward) and 5′-GTAAATCA-CACGGCGCTCTT-3′ (reverse); *Igfbp1*, 5′-TGGTCAGGGAGCCTGTGTA-3′ (forward) and 5′-ACAGCAGCCTTTGCCTCTT-3′ (reverse); *Agxt2l1*, 5′-CAGCTCGGGCATGGAATA-3′ (forward) and 5′-AGCACAGCCAAGCCAACT-3′ (reverse); *Mmd2*, 5′-CTGTGCCACCCATGCTTT-3′ (forward) and 5′-GTCGTCATCGGACAGGAAGT-3′ (reverse); *Pck1*, 5′-ATGTGTGGGCGATGA-CATT-3′ (forward) and 5′-AACCCGTTTTCTGGGTTGAT-3′ (reverse); *G6pc*, 5′-TCTGTCCCGGATCTACCTTG-3′ (forward) and 5′-GAAAGTTTCAGCCA-CAGCAA-3′ (reverse).

**Microarray analysis**. The cDNA was purified and fragmented. We checked for fragmentation with a 2100 Bioanalyzer, and the cDNA was then end-labeled with 153 biotin, using terminal transferase (WT terminal labeling kit, Affymetrix). The cDNA was then hybridized to GeneChip® Mouse Gene 2.0 ST Arrays (Affymetrix) at 45 °C for 17 h. The chips were washed on an FS450 fluidic station FS450, according to specific protocols, and scanned with a GCS3000 7G. The image was analyzed with Expression Console software to obtain raw data (CEL files) and metrics for quality control. The data obtained have been deposited in the Gene Expression Omnibus (GEO) database. Microarray data were analyzed with R-based BRB-Array Tools, as described in Just et al.[21].

**Mass spectrometry (MS) for label free protein quantification (LFQ) and phosphoproteomics**. We grounded and homogenized 50 mg of liver tissue in 500 μl Tris/SDS buffer (Tris/SDS buffer: 50 mM Tris/HCl, pH 8.5, 2% SDS) in an Ultra Turrax apparatus. The homogenate was incubated for 5 min at 95 °C, and centrifuged at 20,000 × *g* for 15 min. The protein concentration of the supernatant was determined in a bicinchoninic acid assay (BCA, Pierce). We reduced and alkylated 50 μg of protein with 20 mM TCEP (tris(2-carboxyethyl)phosphine) and 50 mM chloroacetamide

(both from Sigma) and digested the products with trypsin (sequencing grade from Promega), by the FASP method[54]. Peptides were cleaned and fractionated with $C_{18}$ and SCX StageTips, respectively as previously described[54]. MS analysis of the peptides was performed with a nano liquid chromatographer (nLC) hyphenated to a Q-Exactive Plus mass spectrometer (both from Thermo Electron). Peptides corresponding to one µg of digested proteins was injected for each fraction. The MS acquisition was performed according to a data-dependent scheme to analyze the nLC elution content by electrospray ionization as follows: SCX StageTip peptide fractions were separated on a reverse-phase column (Pepmap $C_{18}$ 2 µm particle size, 100 Å pore size, 75 µm inner diameter, 15 cm length from Thermo) with a 3-h gradient starting with 99% of solvent A containing 0.1% formic acid in $H_2O$ and ending with 40% of solvent B containing 80% ACN and 0.085% formic acid in $H_2O$. The mass spectrometer acquired data throughout the elution process. The MS scans spanned from 350 to 1500 Th with automated gain control (AGC) target at $1 \times 10^6$, within 60 ms maximum ion injection time (MIIT) and a resolution of 70,000. Higher energy Collisional Dissociation (HCD) fragmentation was performed on the 10 most abundant ions, with a dynamic exclusion time of 30 s. The precursor selection window was set at 2Th. The HCD normalized collision energy (NCE) was set at 27% and MS/MS scan resolution was set at 17,500, with AGC target $1 \times 10^5$ within 60 ms MIIT. Spectra were recorded in profile mode. The mass spectrometry data were analyzed with Maxquant version 1.5.2.8[55]. The database used was a concatenation of murine sequences from the Uniprot-Swissprot database (Uniprot, release 2015-02) and the list of contaminant sequences from Maxquant. Cysteine carbamidomethylation was set as a constant modification and acetylation of the protein N-terminus and methionine oxidation were set as variable modifications. Second peptide search and the "match between runs" (MBR) options were allowed. The false discovery rate (FDR) was kept below 1% for both peptides and proteins. Label-free protein quantification (LFQ) was performed with both unique and razor peptides, and required at least two such peptides. Statistical analysis and data comparisons were performed with Perseus software[56].

*Phosphoproteome analysis.* Phosphopeptides were purified according to Humphrey et al[57]. Briefly, livers were minced and homogenized in homogenization buffer (100 mM Tris/HCl, pH 8.00 containing 6 M guanidinium chloride). Proteins were quantified and 3 mg of proteins were reduced and alkylated by heating to 95 °C for 5 min with 10 mM TCEP and 40 mM chloroacetamide. Proteins were precipitated with cold acetone, resolubilized in 100 mM bicarbonate ammonium solution containing 10% trifluoroethanol and digested overnight with 60 µg trypsin. KCl, $KH_2PO_4$, ACN and TFA were added to final concentrations of 300 mM, 5 mM, 50%, and 6%, respectively. Samples were centrifuged at $20,000 \times g$ for 10 min at room temperature and transferred to 30 mg $TiO_2$ bead pellets. Samples with $TiO_2$ beads were incubated for 5 min at 40 °C with continuous agitation, beads were recovered by centrifugation, washed with 50% ACN containing 1%TFA and transferred to C8 StageTips in 80%ACN containing 0.5% acetic acid. Peptides were eluted with 50% ACN containing 15% $NH_4OH$ and immediately acidified by TFA. Peptides were dried and cleaned on SDB-RPS StageTips before analysis by mass spectrometry on a Q-exactive plus as described above, for refed mice, but with the following modifications: The nLC gradient lasted for 2 h, and in the MS mode the scans spanned from 375 to 1500 Th with AGC target $3 \times 10^6$, within 100 ms MIIT. the precursor selection window was set at 4Th. The HCD NCE was set at 27% stepped at 32%. The MS/MS scan resolution was set at 17,500, with AGC target within 100 ms MIIT.

An Orbitrap Fusion mass spectrometer acquired data from fasting mice samples throughout the elution process and operated in a data-dependent scheme with full MS scans acquired with the orbitrap, followed by stepped HCD MS/MS (top speed mode in 3 s) on the most abundant ions detected in the MS scan. MS1 settings were: full MS: AGC at $4 \times 10^5$, resolution: 120,000, *m/z* range 350–1500, within 60 ms MIIT. MS/MS were performed on precursors over $1 \times 10^4$ intensity, with a charge state of 2–4 and with *m/z* quadrupole-filtering window within 1.6 Th. A dynamic exclusion time was set at 30 s. Stepped HCD was performed with NCE at 30 plus and minus 5%, fragments were detected in the ion trap with AGC Target at $1 \times 10^4$ and within 100 ms MIIT. Phopshopeptide data were analyzed with Maxquant version 1.6.6.0[55]. The database used was a concatenation of murine sequences from the reviewed Uniprot-Swissprot database (UniprotKB, release 2019-10) and the list of contaminant sequences from Maxquant. Cysteine carbamidomethylation was set as a constant modification while phosphorylations of STY, acetylation of the protein N-terminus and methionine oxidation were set as variable modifications. Second peptide search and the "match between runs" (MBR) options were allowed. The false discovery rate (FDR) was kept below 1% for both peptides and proteins. Label-free protein quantification (LFQ) was performed with razor peptides, and required at least two such peptides. Statistical analysis and data comparisons were performed with Perseus software[56].

**Production of glucose in primary cultures of hepatocytes.** Primary hepatocytes were isolated from Lkb1KO[livad] mice and their corresponding controls, 15 days after tamoxifen injection, by two-step collagenase perfusion followed by filtration through a 70 µm mesh. Viability was assessed by trypan blue exclusion, and the cells were then plated in William's E medium (supplemented with 10% fetal bovine serum, penicillin–streptomycin and 1% (w/v) bovine serum albumin). Four hours after plating, the medium was replaced with William's E medium containing $10^{-9}$ M dexamethasone and penicillin–streptomycin. Glucose production by primary hepatocytes was measured by incubating the cells for 3 h in glucose-free

DMEM medium containing $10^{-9}$ M dexamethasone, penicillin–streptomycin, and either 2 mM sodium pyruvate and 20 mM sodium lactate, or one of the amino acids tested (Ala, Gln or Ser) at a concentration of 10 mM.

The concentration of glucose in the culture medium at the end of the experiment was assessed with the Glucose GO assay kit (Sigma, USA) according to the manufacturer's instructions.

**Glucose and glycogen synthesis from [U-$^{14}$C]alanine.** Metabolic flux experiments were performed on liver explants (150–200 mg samples in triplicate) in suspension in William's E medium in conical glass vials. Fasted mice were killed and their livers removed. The liver was rapidly chopped into ~2 mm³ pieces and explants were incubated in the presence of 10 mM [U-$^{14}$C]alanine in glucose-free medium for 3 h. At the end of the experiment, the culture medium was recovered for assays of [$^{14}$C] incorporation into glucose, and liver explants were washed three times in ice-cold PBS and processed for the measurement of $^{14}$C incorporation into glycogen. Liver glycogen was extracted from liver explants by homogenizing the tissue by incubation in 5 M KOH at 100 °C at 30 min, followed by two rounds of precipitation in alcohol. $^{14}$C incorporation into the glycogen fraction was then accessed by scintillation counting. The culture medium was deproteinized and [$^{14}$C]glucose was separated from the [$^{14}$C]alanine precursor by ion exchange chromatography, as described by Kreisber et al.[58].

**Determination of amino acid content in the serum, liver, and skeletal muscle.** For amino acid determinations, tissue samples were homogenized in ice-cold 10% TCA-0.5 mM EDTA-250 µM AEC, and plasma samples were deproteinized with 10% sulfosalicylic acid. Amino acids were separated and quantified by ion exchange chromatography with post-column ninhydrin derivatization, with a JLC-500/V AminoTac amino acid analyzer (Jeol, Croissy sur Seine, France).

**Software analyses.** Proteomic analyses were done using MaxQuant (v1.6.6.0) and Perseus softwares (v1.6.6.0).

The functional analyses of both proteomic and phosphoproteomic data were generated through the use of IPA (QIAGEN Inc., https://www.qiagenbioinformatics.com/products/ingenuity-pathway-analysis, v:49932394, Release Date: 2019-11-14). Enrichments were performed using overrepresentation analysis validated by a right tailed Fisher's exact test. The RBP list was added to the IPA database as a new term.

To construct the PPI network in the STRING database, *Mus musculus* was selected as the organism and the option of multiple proteins was selected. Interaction combined score _0.4 was regarded as significant.

Heatmap were done using the genesis software (https://genome.tugraz.at).

Venn diagram were performed with FunRich (v3.1.3) or using the website http://www.pangloss.com/seidel/Protocols/venn.cgi.

Multigauge V3.0 (Fujifilm) for quantification of western blot.

**Statistical analysis.** Statistical analysis was performed with GraphPad prism (7.04). Graphical data are represented as means ± SD in most of the cases. Unpaired two-tailed Student's *t* tests were used to assess differences between two groups. Values of *p* below 0.05 were considered significant. *$p < 0.05$, **$p < 0.001$, ***$p < 0.001$, ****$p < 0.0001$.

For indirect calorimetry, the results are expressed as mean ± SEM. Indirect calorimetry data presented are the mean of at least 96 h of measurement.

**Reporting summary.** Further information on research design is available in the Nature Research Reporting Summary linked to this article.

## Data availability

The proteomic data were obtained using the Uniprot-Swissprot database (Uniprot, release 2015-02). All proteomic data that support the finding of this study have been deposited in the repository PRIDE with the accession numbers PXD013478, PXD019757, and PXD019755. The sequencing data that support the finding of this study have been deposited in the the Gene expression Ommnibus with the accession number GSE75564 already described in[21] and with the accession number GSE132536. Uncropped scans of Western blots are included in the Source Data file. Source data are provided with this paper.

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

## Acknowledgements

Core fundings came from INSERM and CNRS grants, and grants from the "Société Francophone du Diabète"(SFD) and the Ligue Nationale Contre le cancer (LNCC). LNCC and SFD are non-profit organizations that did not have any support in the form of salaries for authors. The Orbitrap Fusion mass spectrometer was acquired with funds from the FEDER through the "Operational Programme for Competitiveness Factors and employment 2007-2013" and from the "Canceropole Ile de France". We would like to thank Dominique Weil (Institut de Biologie Paris Seine, France) for her fruitful discussion and Marc Billaud, Benoit Viollet, Anne-Francoise Burnol, Pascal Maire (Institut Cochin) for their comments. We would also like to thank Fadila Rayah, Julien Piedcoq, and Dalila Azzout Marniche for insulin and glucagon determinations, Jean-Marc Masse for the Agxt immunogold analysis, Yousra Lottin, Cedric Broussard, and Evangeline Bennana for quality management, mass spectrometers management, and sample preparation, respectively.

We also acknowledge the animal core facility of Institut Cochin and thank Nadia Boussetta for taking care of the mice, the animal core facility 'Buffon' of the University of Paris/Institut Jacques Monod, for animal husbandry and care and thank Le Parco Isabelle, Angélique Dauvin and Daniel Quintas for care of animals. We thank the Functional and Physiological Exploration platform of the BFA Unit for metabolic analysis.

## Author contributions

C.P., P.A.J., S.C., and P.B. conceived and performed experiments. C.P. and B.R. secured funding. C.P., P.B., and J.P.D.B., wrote the manuscript. R.G.P.D. and S.L. performed experiments for MRI investigation. M.L., F.G., and P.M. performed experiments for proteomic analysis. M.L.G. and C.P. performed ontology and data analyses, J.P.D.B., and S.M. performed the amino acid determinations. M.S., M.T., S.B., N.S., P.S., M.W., M.F., and A.S. performed experiments. M.V.C. provided the TTR-CreTam mice. All the authors approved the final version for submission.

## Competing interests

The authors declare no competing interests.
