## [Peer Review File · Nature Communications]

Reviewers' comments:

Reviewer #1 (Remarks to the Author):

The present manuscript examines the role of liver amino acid degradation in hyperglycemia induced by LKB1 loss. Specifically, it shows that LKB1 suppresses gluconeogenesis in part by blocking amino acid degradation. Strikingly, it shows that liver LKB1 loss causes systemic muscle wasting, a striking observation of substantial physiological significance. Although I am not an LKB1 or gluconeogenesis expert, overall, I found this to be an interesting, clear, and objective manuscript.

While hypothetically one could ask for isotope tracing studies to prove an increased amino acid contribution to circulating glucose in the LKB1 KO mice, I found the paper adequately persuasive without this. From my perspective, such experiments are worthwhile but can be left to other experts or a collaborative study in the future.

Accordingly, I have only minor comments:

1. Fig 1g, Y-axis should be marked "days"
2. Why are there so many grey boxes in Fig 3f?
3. I would remove "critical" from the last sentence of the abstract.
4. The authors should be very specific about the reactions of Agxt and how these might relate to gluconeogenesis (which is not so obvious) (probably include diagram).

Overall, high quality work of scientific significance.

Reviewer #2 (Remarks to the Author):

Summary

The current manuscript by Just et al. explores the role of liver kinase B1 (LKB1) in amino acid-dependent regulation of hepatic gluconeogenesis. By using liver-specific LKB1 KO mice, the authors demonstrate that the hepatic loss of LKB1 led to hyperglycemia and –upon aging- to a loss in body weight and skeletal muscle mass. Overall, energy expenditure was found to be unaltered but body composition shifted from skeletal muscle to fat mass after LKB1 deletion. Proteomics analysis identified amino acid catabolism as one of the most significantly induced pathways upon LKB1 KO. In line with these findings, LKB1 KO animals were able to more efficiently produce glucose from alanine as compared to WT littermates, correlating with amino acid deficiency in skeletal muscle. Finally, amino transferase inhibition was found to rescue the LKB1-driven gluconeogenic phenotype, and deficiency in alanine glyoxylate aminotransferase (Agxt) partially rescued the LKB1-dependent glucose phenotype. Overall, the authors conclude that LKB1 is a suppressor of amino acid-driven hepatic gluconeogenesis and might represent a target for anti-hyperglycemic drugs in the treatment of type 2 diabetes.

General comments

Excessive hepatic gluconeogenesis represents a key driver for hyperglycemia in both type 1 and type 2 diabetes. Also, it fulfills a critical role in the maintenance of glucose homeostasis during the daily

fasting-feeding cycle. In this context, the current manuscript by Just et al. covers an interesting and important topic. By using a combination of KO mouse models, primary cells and high throughput technology the authors convincingly demonstrate a role for LKB1 in amino acid-driven gluconeogenesis and also document a potentially interesting liver-muscle cross-talk mechanism. The conclusions are supported by the experimental data and the manuscript is well-structured and written. However, two main issues still require attention by the authors: a) The mechanistic link between LKB1 signaling and the regulation of hepatic aminotransferases remains completely unclear. In order to address this issue, the authors should perform a phospho-proteomics analysis from WT and LKB KO animals to identify potentially involved mechanisms/pathways controlled by post-translational modifications that could explain the regulatory impact of LKB1 on amino acid metabolism in the liver (e.g. is AMPK involved?). b) As stated by the authors, aberrant hepatic gluconeogenesis is a key driver of hyperglycemia in diabetes. At this stage, the current manuscript does not explore the importance of the LKB1 pathway in this setting: Can you prevent glucose intolerance and insulin resistance induced by high fat diet feeding in LKB1 KO mice? Can you reverse hyperglycemia in db/db mice by inactivating LKB1, e.g. by using an AAV-driven hepatic knock down? To which degree is the LKB1-dependent amino acid target gene network dysregulated under diabetic conditions, and to which degree does the LKB1 pathway contribute to diabetic gluconeogenesis driven by amino acid catabolism? These issues seem critical to increase the conceptual advance of the current study as compared with long-known established functions of LKB1 in gluconeogenesis. Specific comments in this direction are listed below.

Specific comments

1. Fig.1: What is the basis for the loss of skeletal muscle: Please provide data on catabolic pathways in muscle, e.g. the expression of ubiquitin ligases and insulin signaling strength.
2. Fig.2: Is there fecal energy loss?
3. Fig.6: Please provide a more detailed metabolic analysis of single and DKO mice regarding glucose tolerance, insulin and lipid levels. Are energy expenditure and food intake altered in Agxt KO/DKO mice?

Reviewer #3 (Remarks to the Author):

The manuscript entitled “Lkb1 suppresses amino acid-driven gluconeogenesis in the liver” by C. Perret and colleagues uncovers new mechanisms of LKB1-dependent suppression of gluconeogenesis. Interestingly, the authors identified that LKB1 controls gluconeogenesis through transaminases, and more specifically highlighted AGXT as one of the enzymes involved in this process. They conducted a solid and straightforward study based on several mouse models of adult and embryonic Lkb1 inactivation and a double knockout model that partially rescues the phenotype, mice phenotyping using calorimetry cages and MRI, transcriptomics and proteomics, and several approaches to evaluate cell metabolism performed in vivo, on liver explants or in primary hepatocyte cultures: tolerance test to several metabolites, flux analyses and metabolite dosages (in particular whole amino acids profiling) from several tissues. Overall, this thorough description demonstrates that LKB1 controls transaminases, specifically AGXT to maintain hepatic and blood glucose levels and opens new perspectives of therapeutic investigations especially for diabetic

patients. However, there are some concerns that should be addressed.

Major concerns:

- 1- The relationship between glycogen levels and transaminases activity has not been explored although it is an interesting point that could well complement the authors' observations. First AMPK has been shown to directly regulate glycogen synthase enzyme Gys2 by an inhibitory phosphorylation on Ser7 in human Gys2. Study of Gys2 phosphorylation status is critically missing in Fig1f. Second, are glycogen levels also rescued when transaminases are inhibited with AOA or in the double knockout mice? Using flux analyses, the authors nicely observed that glycogen accumulated from alanine. There could be thus two independent ways for LKB1 to control glucose storage into glycogen (one through AMPK activity on Gys2, one through transaminases activity) and the authors could include comments on this dual regulation of glycogen pool in the discussion.
- 2- Along this line, it would be interesting to evaluate if AMPK contributes at least in part in transaminases and consequently gluconeogenesis regulation downstream of LKB1 in hepatocytes.
- 3- AGXT biochemical reactions (alanine-glyoxylate aminotransferase EC2.6.1.44 and serine-pyruvate aminotransferase EC2.6.1.51) should be clarify in the text and schematized in Fig6. There are at least two isoforms of AGXT one being peroxisome associated and probably more involved in glyoxylate detoxification. Assessing AGXT localization in hepatocytes and an eventual relocalization in subcellular compartments upon Lkb1 loss would be interesting. The authors identified one AGXT isoform as a direct target of Foxo-1, a master transcriptional activator of gluconeogenesis but surprisingly, they did not comment further this information. Including discussion about the possible link with Foxo-1, AGXT isoforms as well as which transaminase activity could be involved in gluconeogenesis suppression would bring value to the discussion.
- 4- A general schema summarizing the authors' findings including both transaminases and glycogen should be included.

Minor concerns:

- 1- L101 PAS staining is mentioned in the text to stain glycogen. However, PAS stains wider molecules such as polysaccharides, glycoproteins and glycolipids, glycogen being only one of them although it is the most abundant.
- 2- Data not shown L99 & L117 especially regarding liver histology could be included in sup data.
- 3- L181 correct "the critical enzymes... showed a strong upregulation of their expression" by "... of their protein levels..."
- 4- L122 & L254 "invalidation" should be replace by "inactivation"
- 5- L228 "through its control"
- 6- L247 consequences of Lkb1 inactivation during embryonic stages on glucose levels are not mentioned. Such information would bring value to the article and enhance the interest of the authors' transcriptomic data.
- 7- L257 "Fig. 5e" should be corrected by "Fig. 6e".
- 8- L512 the sentence is not correct ; probably add "Glucagon was determined by immunoassay..."
- 9- Scale bars are missing or not visible fig 1e
- 10- Fig 1g time scale (days) is not indicated
- 11- Fig1h: white arrows are difficult to find (color can be changed). It is hard to see the adipocyte tissue, maybe a zoom could improve this. Indicate the age of the individuals in the legend. Both

decreased adipocyte tissue (Fig1h) and increased fat mass (Fig2C) are described; the explanation probably relies on mouse ages between both analyses and should be further commented in the manuscript.

12- On graph fig4a, stars indicative of the statistical significance have to be moved up.

13- Quantifications of results shown Fig5a would be helpful.

14- Standard errors should be added in Fig S1f and scale bars are missing.

15- Legend of the graph in SupFig4c is not correct or enough detailed or does not fit with the text legend.

16- Supplementary legends could be more detailed

Point-by-point response to « Lkb1 suppresses amino acid-driven gluconeogenesis in the liver” by Just PA et al.,

The comments of the reviewer are in black italics, and the response in blue.

Reviewer #1

The present manuscript examines the role of liver amino acid degradation in hyperglycemia induced by LKB1 loss. Specifically, it shows that LKB1 suppresses gluconeogenesis in part by blocking amino acid degradation. Strikingly, it shows that liver LKB1 loss causes systemic muscle wasting, a striking observation of substantial physiological significance. Although I am not an LKB1 or gluconeogenesis expert, overall, I found this to be an interesting, clear, and objective manuscript.

While hypothetically one could ask for isotope tracing studies to prove an increased amino acid contribution to circulating glucose in the LKB1 KO mice, I found the paper adequately persuasive without this. From my perspective, such experiments are worthwhile but can be left to other experts or a collaborative study in the future.

First of all, we thank the reviewer for the constructive and positive comments and for the finding that the manuscript is persuasive without isotope tracing studies..

Accordingly, I have only minor comments:

1. Fig 1g, Y-axis should be marked "days"

Thank you for highlighting this error. It has now been corrected.

.

2. Why are there so many grey boxes in Fig 3f?

The grey boxes in Fig3f indicates that the proteins in gray were not quantified (designated NaN in Table S1 and S2). Two reasons may explain this, (1) it is linked to the MS/MS method and is related to the sequencing speed ¹, (2) the quantity of the protein is too low to allow the identification of peptides. Data presented on Fig.3f indicates that there were many more proteins of the amino acid metabolism that were differentially expressed (up-regulation) between KO and controls in the refed state than in the fasted state. This data supports the hypothesis that Lkb1 controls the post-prandial neogluconeogenesis, known to allow the stockage of the diet amino-acids from the meal to be stocked as glycogen.

3. I would remove "critical" from the last sentence of the abstract.

We agree with the comment and deleted the word « critical » of the abstract.

4. The authors should be very specific about the reactions of Agxt and how these might relate to gluconeogenesis (which is not so obvious) (probably include diagram).

We thank the reviewer for this comment that has helped us to better explain the functions of Agxt.

Agxt is an enzyme with two dual metabolic functions of either gluconeogenesis in mitochondria or glyoxylate detoxifications in both peroxisomes and mitochondria. Depending of the species and likely linked to the diet, Agxt can be found either in the peroxisomes or mitochondria, or in both peroxisomes and mitochondria. This variable localization depends on the differential expression of N-terminal mitochondrial and C-terminal peroxisomal localization sequences thanks to alternative transcription and translation initiation sites. In rodents, Agxt is both mitochondrial and peroxisomal ². Interestingly, in rats, when fed on high protein diets or submitted to a gluconeogenic stimuli, such as glucagon, there is a large increase in the synthesis of mitochondrial, but not peroxisomal Agxt ³. We thus analyzed if Lkb1 controls the distribution of Agxt by performing colloid gold immunoelectron microscopy on liver of *Lkb1*-deficient and control mice. The quantification of gold particles in mitochondria and peroxysome showed an increase in Agxt in both organelles in mutant animals compared to controls (**Fig 7e, Supplementary Fig. 5d**), slightly higher in KO vs. WT and in mitochondria compared to peroxisome. These results suggested that Lkb1 controls the targeting of Agxt in both the mitochondria and the peroxisome without leading to a relocalization of Agxt in mitochondria, the site of gluconeogenesis.

These new results have been added in Result section (**Fig 7e, Supplementary Fig. 5d**), and described in the Result section, **p15**.

We thank the reviewer for the comments concerning a better description of the function of Agxt. We added a scheme (**Fig. 6c**) and a comment on its dual function (**p15**).

Overall, high quality work of scientific significance.

Reviewer #2

Summary

The current manuscript by Just et al. explores the role of liver kinase B1 (LKB1) in amino acid-dependent regulation of hepatic gluconeogenesis. By using liver-specific LKB1 KO mice, the authors demonstrate that the hepatic loss of LKB1 led to hyperglycemia and –upon aging- to a loss in body weight and skeletal muscle mass. Overall, energy expenditure was found to be unaltered but body composition shifted from skeletal muscle to fat mass after LKB1 deletion. Proteomics analysis identified amino acid catabolism as one of the most significantly induced pathways upon LKB1 KO. In line with these findings, LKB1 KO animals were able to more efficiently produce glucose from alanine as compared to WT littermates, correlating with amino acid deficiency in skeletal muscle. Finally, amino transferase inhibition was found to rescue the LKB1-driven gluconeogenic phenotype, and deficiency in alanine glyoxylate aminotransferase (Agxt) partially rescued the LKB1-dependent glucose phenotype.

Overall, the authors conclude that LKB1 is a suppressor of amino acid-driven hepatic gluconeogenesis and might represent a target for anti-hyperglycemic drugs in the treatment of type 2 diabetes.

General comments

Excessive hepatic gluconeogenesis represents a key driver for hyperglycemia in both type 1 and type 2 diabetes. Also, it fulfills a critical role in the maintenance of glucose homeostasis during the daily fasting-feeding cycle. In this context, the current manuscript by Just et al. covers an interesting and important topic. By using a combination of KO mouse models, primary cells and high throughput technology the authors convincingly demonstrate a role for LKB1 in amino acid-driven gluconeogenesis and also document a potentially interesting liver-muscle cross-talk mechanism. The conclusions are supported by the experimental data and the manuscript is well-structured and written. However, two main issues still require attention by the authors: a) The mechanistic link between LKB1 signaling and the regulation of hepatic aminotransferases remains completely unclear. In order to address this issue, the authors should perform a phospho-proteomics analysis from WT and LKB KO animals to identify potentially involved mechanisms/pathways controlled by post-translational modifications that could explain the regulatory impact of LKB1 on amino acid metabolism in the liver (e.g. is AMPK involved?).

We thank the reviewer for his positive and constructive comments. To address the involvement of Ampk, we used mice deficient in *Ampk* in the hepatocytes. By western blot we found that the level of several proteins of the amino acid catabolism did not change between the mutant and the controls. This result indicates that Lkb1 acts independently of Ampk.

We would like to thank the reviewer for the suggestion to perform a comprehensive phosphoproteomic analysis which has helped to improve our manuscript. We have performed quantitative phosphoproteomic analysis on liver of mutant and control mice both in the fasted and refed state. The *in silico* analyses of our data led us to identify potential clues that may explain how Lkb1 could control at the translational level the expression of proteins of the amino acid metabolism. Indeed, we identified an enrichment of RNA binding proteins that were differentially phosphorylated between the mutant and control mice. These proteins are key actors of the messenger ribonucleoprotein (mRNP) complexes which are highly dynamic and for which phosphorylation is a key regulatory mechanism and participates to the control of mRNA translation⁴. Interestingly, our data suggest that the mechanism by which Lkb1 controls hepatic amino acid-driven gluconeogenesis differs between the fasted and refed states.

All these data that highlighted the mechanism of action of Lkb1 in hepatic amino acid driven gluconeogenesis have been added in the new **Fig.7** and Supplementary Fig.5 and have been described in the **Result Section p 12-15**, and discussed in the **Discussion section p 18**.

b) As stated by the authors, aberrant hepatic gluconeogenesis is a key driver of hyperglycemia in diabetes. At this stage, the current manuscript does not explore the importance of the LKB1 pathway in this setting: Can you prevent glucose intolerance and insulin resistance induced by high fat diet feeding in LKB1 KO mice? Can you reverse hyperglycemia in db/db mice by inactivating LKB1, e.g. by using an AAV-driven hepatic knock down?

Although we agree with the referee that it would be interesting to study the role of Lkb1 in the setting of diabetes, Lkb1 is a suppressor of gluconeogenesis. Its hepatic deletion leads to deregulated hyperglycemia would worsen the phenotype of models of diabetic mice. It has been published in ⁵. An interesting strategy would be to activate the Lkb1 signaling, but to my knowledge, no such activator is yet available.

To which degree is the LKB1-dependent amino acid target gene network dysregulated under diabetic conditions, and to which degree does the LKB1 pathway contribute to diabetic gluconeogenesis driven by amino acid catabolism?

It is a very interesting point and we have indeed searched the literature for works supporting an implication of amino acid catabolism in dysregulated gluconeogenesis during diabetes. Most papers describe elevated circulating levels of the branched-chain amino acids (BCAA: leucine, isoleucine and valine) associated with obesity, insulin resistance and T2DM. This is associated to a defect in the degradation pathway of BCAA likely linked to a defect in hypothalamic insulin signaling ⁶. It is thus unrelated to the Lkb1 network we described in this manuscript.

However, we found a study that describes a reduction in postprandial blood glucose in mice when given a low-carbohydrate, high-protein, and high-omega-3 diet. The authors showed that it was associated with an inhibition of hepatic alanine transaminase and reduction of amino acid gluconeogenesis. In addition, this diet rescued the hyperglycemia phenotype of different models of diabetes ⁷. This study is consistent with our study and we have added this point in the **Discussion section p19**.

These issues seem critical to increase the conceptual advance of the current study as compared with long-known established functions of LKB1 in gluconeogenesis. Specific comments in this direction are listed below.

Specific comments

1. Fig.1: What is the basis for the loss of skeletal muscle: Please provide data on catabolic pathways in muscle, e.g. the expression of ubiquitin ligases and insulin signaling strength.

We thank the reviewer for the suggestion to test the catabolic pathways or variation in insulin signaling in the skeletal muscle. We have done these experiments that are presented in the **Supplementary Fig. 1i, 1j**. We did not find any significant difference that may explain the loss of skeletal muscle. All these data support the hypothesis that the skeletal muscle loss is an indirect consequence of the increase hepatic amino acid extraction for

gluconeogenesis. This has been added in the **Result section (p 7)** and in the **Discussion section (p 16)**.

2. Fig.2: Is there fecal energy loss?

We have tested the hypothesis of a fecal energy loss, but as shown on this graph, we did not find any significant difference between the mutant and control animals.

3. Fig.6: Please provide a more detailed metabolic analysis of single and DKO mice regarding glucose tolerance, insulin and lipid levels. Are energy expenditure and food intake altered in Agxt KO/DKO mice?

We apologize for the minimal metabolic analyses of the DKO mice. We have done GTT, measured glycogen content as well as plasma insulin and lipid levels. We also analyzed energy expenditure and food intake. All these data are presented in the **Supplementary Fig. 4c, 4d, 4e, 4f, 4g, 4h, 4i**).

We have not found major changes in the DKO in the metabolic analyses as well as the energy expenditure and food intake compared to the KO mice. Specifically, we did not see any change in the glycogen content suggesting that even if Agxt is a key effector of the Lkb1-dependent amino acid-driven gluconeogenesis, its sole deletion is not sufficient to rescue the glycogen phenotype of Lkb1KO^{livad} mice. We have added this comment in the **Result section p 12**. However, Agxt appears to have a specific role, since its deletion partially rescued the cachectic phenotype. We have also deleted the term « key » in the abstract and introduction when presenting Agxt as effector of the Lkb1 signaling.

We are sincerely grateful for all the constructive comments.

Reviewer #3

The manuscript entitled “Lkb1 suppresses amino acid-driven gluconeogenesis in the liver” by C. Perret and colleagues uncovers new mechanisms of LKB1-dependent suppression of gluconeogenesis. Interestingly, the authors identified that LKB1 controls gluconeogenesis through transaminases, and more specifically highlighted AGXT as one of the enzymes involved in this process. They conducted a solid and straightforward study based on several mouse models of adult and embryonic Lkb1 inactivation and a double knockout model that partially rescues the phenotype, mice phenotyping using calorimetry cages and MRI, transcriptomics and proteomics, and several approaches to evaluate cell metabolism performed in vivo, on liver explants or in primary hepatocyte cultures: tolerance test to several metabolites, flux analyses and metabolite dosages (in particular whole amino acids profiling) from several tissues. Overall, this thorough description demonstrates that LKB1 controls transaminases, specifically AGXT to maintain hepatic and blood glucose levels and opens new perspectives of therapeutic investigations especially for diabetic patients. However, there are some concerns that should be addressed.

First of all, we thank the reviewer for his constructive and positive comments.

Major concerns:

1- The relationship between glycogen levels and transaminases activity has not been explored although it is an interesting point that could well complement the authors' observations. First AMPK has been shown to directly regulate glycogen synthase enzyme Gys2 by an inhibitory phosphorylation on Ser7 in human Gys2. Study of Gys2 phosphorylation status is critically missing in Fig1f. Second, are glycogen levels also rescued when transaminases are inhibited with AOA or in the double knockout mice? Using flux analyses, the authors nicely observed that glycogen accumulated from alanine. There could be thus two independent ways for LKB1 to control glucose storage into glycogen (one through AMPK activity on Gys2, one through transaminases activity) and the authors could include comments on this dual regulation of glycogen pool in the discussion.

We thank the reviewer for his relevant remark concerning the inhibitory regulation of Gys2 by Ampk. We have done a comprehensive phosphoproteomic analysis of the liver of Lkb1 mutant and control animals that is presented in Figure 7. The differentially expressed (DE) phosphopeptides between the mutant and control liver are listed on **Supplementary**

Table S4. As shown on this table, a significant difference for phosphorylated Gys2 has been found only in the group « Refed », for the phosphopeptide around Ser11. It could potentially correspond to a phosphorylation by Ampk. However, our data showed an increase in the phosphorylated Ser11-Gys2 ratio between KO and controls. Since Ampk is a target of Lkb1, we should expect the opposite, that is a decrease in the phosphorylated Ser11 between KO and controls. Furthermore, this phosphorylation is inhibitory. We did not find any differentially phosphorylated Gys2 in the fasted state.

Altogether, these data do not support that the activity of Gys2 may be controlled by a phosphorylation event, and is likely due solely to the increase in Gys2 protein level. This has been added in the **discussion p18**.

We also analyzed the hepatic glycogen level in the DKO mice, but did not find any rescue effect of the deletion of Agxt on the glycogen content, suggesting that even if Agxt is a key effector of the Lkb1-dependent amino acid-driven gluconeogenesis, its sole deletion is not sufficient to rescue the glycogen phenotype of Lkb1KO^{liv^{ad}} mice. This data have been described in the **Result section (p12)** and data have been added in the new **Fig.7e** and in **Supplementary Fig.5d**. However, Agxt appears to have a specific role, since its deletion partially rescued the cachectic phenotype.

2- Along this line, it would be interesting to evaluate if AMPK contributes at least in part in transaminases and consequently gluconeogenesis regulation downstream of LKB1 in hepatocytes.

To address if AMPK contributes to the controls of transaminases by Lkb1 in the liver, we performed WB using liver of mice deficient for both *Ampka1* and *Ampka2* in the hepatocytes⁸. The western blot clearly showed that Ampk did not control the expression of the transaminases Got1 and Agxt, or Oat, strongly suggesting that Lkb1 controls the amino-acid metabolism independently of Ampk.

This has been added in the text of the **Result section p 12** in the **Fig.7a**.

3- AGXT biochemical reactions (alanine-glyoxylate aminotransferase EC2.6.1.44 and serine-pyruvate aminotransferase EC2.6.1.51) should be clarify in the text and schematized in Fig6. There are at least two isoforms of AGXT one being peroxisome associated and probably more involved in glyoxylate detoxification. Assessing AGXT localization in hepatocytes and an eventual relocalization in subcellular compartments upon Lkb1 loss would be interesting. The authors identified one AGXT isoform as a direct target of Foxo-1, a master transcriptional activator of gluconeogenesis but surprisingly, they did not comment further this information. Including discussion about the possible link with Foxo-1, AGXT isoforms as well as which transaminase activity could be involved in gluconeogenesis suppression would bring value to the discussion.

We thank the reviewer for his comment about distribution and function of Agxt and have studied the distribution of Agxt in the liver of *Lkb1*-deficient mice compared to controls. It is an interesting point since, Agxt is an enzyme with two dual metabolic functions of gluconeogenesis in mitochondria and glyoxylate detoxifications in peroxisomes. Depending of the species and likely linked to the diet, Agxt can be found either in the peroxysomes or mitochondria, or in both peroxisomes and mitochondria. In rodents, Agxt is both

mitochondrial and peroxisomal ¹. We have added a scheme in **Fig. 6c** that described the dual metabolic function of Agxt.

Interestingly, in rats, when fed on high protein diets or given gluconeogenic stimuli, such as glucagon, there is a large increase in the synthesis of mitochondrial, but not peroxisomal Agxt ³. We thus analyzed if Lkb1 controls the distribution of Agxt by performing colloid gold immunoelectron microscopy on liver of *Lkb1*-deficient and control mice. Our results are shown on the **Fig. 7e** and showed the quantification of gold particules in each organelle (mitochondria or peroxisome). We observed an increase in Agxt between the KO and WT in both organelles, although with a slightly larger increase for the mitochondria compared to the peroxysome. These results suggested that Lkb1 controls the targeting of Agxt in both the mitochondria and the peroxysome without leading a relocalization of Agxt in mitochondria, the site of gluconeogenesis.

These new results have been added in (**Fig 7e, Supplementary Fig. 5d**), and described in the result section, **p15**.

Concerning the control of Agxt by Foxo1 suggested by the reviewer. We are not sure that Agxt is controlled by Foxo-1. The result we presented in Fig.3b corresponds to another gene, Agxt2L1 that is located on another chromosome (In human, *AGXT2L1* is on 4q25, while *AGXT* is on 2q27). Furthermore, the function of Agxt2L1 is not well characterized as it is for Agxt. Agxt2L1 has been described as a Foxo1 target in a previous study ⁹.

4- A general schema summarizing the authors' findings including both transaminases and glycogen should be included.

We thank the reviewer for his suggestion to add a general schema showing how Lkb1 negatively controlled the amino acid catabolism including the aminotransferases to provide amino acid precursors for the synthesis of glucose and then glycogen. This has been added in **Fig.7f**.

Minor concerns:

1- L101 PAS staining is mentioned in the text to stain glycogen. However, PAS stains wider molecules such as polysaccharides, glycoproteins and glycolipids, glycogen being only one of them although it is the most abundant.

We agree with the comment of the reviewer that PAS staining do not always identify glycogen, but since we also have measured the glycogen content by enzymatic analysis, we are confident that there is an increase in glycogen content in the liver of fasted mutant mice compared to controls.

We modified the text accordingly (**p 5**). 'However, the PAS staining of liver sections **associated with** the determination of glycogen content revealed a much higher level of glycogen accumulation in mutant than in control mice, in the fasted state.

2- Data not shown L99 & L117 especially regarding liver histology could be included in sup data.

We have deleted the data not shown and added the H&E staining of WT and KO animals in **Supplementary Fig1g**.

3- L181 correct “the critical enzymes... showed a strong upregulation of their expression” by “... of their protein levels...”

4- L122 & L254 “invalidation” should be replace by “inactivation”

5- L228 ”through its control”

Thank you for highlighting these errors. They have now been corrected.

6- L247 consequences of *Lkb1* inactivation during embryonic stages on glucose levels are not mentioned. Such information would bring value to the article and enhance the interest of the authors’ transcriptomic data.

It is an interesting point, but the model of *Lkb1* inactivation during liver embryonic stage (*Lkb1*^{fl/fl}, AIFP-cre) led to a strong developmental phenotype with cholestasis that we previously published¹⁰. Mutant mice showed a large decrease of bile ducts which were all abnormal. In addition, the mutant mice revealed a major growth retardation that began at around 12 days post-birth (see figure below), most of the mutant mice die before 1 month.

We have measured the glycemia of these mutant mice (see figure below). The glycemia was normal until 22 days post-birth, and then we observed a significant hypoglycemia. However, given the complexe developmental phenotype of these mice, it is difficult to compare this data with the one we described in the present manuscript where *Lkb1* is invalidated in the adult hepatocytes. In our opinion, adding these data to the manuscript would bring complexity in the understanding of the phenotype of the *Lkb1*KO^{livad} mice.

Legend : Growth curve of *Lkb1* mutant mice where the deletion of *Lkb1* has been done in embryonic hepatoblast¹⁰.

Legend : Glycemia of Lkb1 mutant mice where the deletion of Lkb1 has been done in embryonic hepatoblast¹⁰. WT, n=4 ; KO, n=4.

- 7- L257 “Fig. 5e” should be corrected by “Fig. 6e”.
- 8- L512 the sentence is not correct ; probably add “Glucagon was determined by immunoassay...”

Thank you for highlighting these errors. They have now been corrected.

- 9- Scale bars are missing or not visible fig 1^e

Thank you for highlighting these errors. They have now been corrected.

- 10- Fig 1g time scale (days) is not indicated

Thank you for highlighting these errors. They have now been corrected.

11- Fig1h: white arrows are difficult to find (color can be changed). It is hard to see the adipocyte tissue, maybe a zoom could improve this. Indicate the age of the individuals in the legend. Both decreased adipocyte tissue (Fig1h) and increased fat mass (Fig2C) are described; the explanation probably relies on mouse ages between both analyses and should be further commented in the manuscript.

We apologize for the lack of clarity of **Fig.1h**. We have improved it and modified the color of the arrows to be more obvious. As pointed by the reviewer, the decreased adipocyte tissue (**Fig. 1h**) and the increase fat mas (**Fig. 2c**) correspond to mice with different age. It corresponds for **Fig.1h** to the acute phenotype observed 15 days after tamoxifen injection in mice that were nearly 2,5 months of age, while **Fig.2c** corresponds to older mice that developed the cachectic phenotype. We have not specified the age of the mice in the legend because the development of

sarcopenia and cachexia varies between 3 and 6 months of age. We only have specified that it is a cachectic mutant.

12- On graph fig4a, stars indicative of the statistical significance have to be moved up.

Thank you for highlighting these errors. They have now been corrected.

13- Quantifications of results shown Fig5a would be helpful.

We agree with the reviewer and have shown quantification of the amino acid levels in the plasma.

It is presented in the new **Fig. 5a**.

14- Standard errors should be added in Fig S1f and scale bars are missing.

The microscopic analysis has been done at 10x magnification. Standard errors cannot be added, since only one WT and one KO was quantified. However, we have realized quantified on three types of skeletal muscle, tibialis, gastrocnemius and soleus. All these modifications have been added in the the **Supplementary Fig. 1h** and its corresponding legend.

15- Legend of the graph in SupFig4c is not correct or enough detailed or does not fit with the text legend.

We thank the reviewer for pointing out this error. Graph Supplementary Fig.4c is identical to graph of Fig.1j and has been deleted.

16- Supplementary legends could be more detailed

We apologize for the lack of details of the legends of the supplementary figures. We have rectified this point on the Supplementary information section.

We are sincerely grateful for all the constructive comments

References

- 1 Michalski, A., Cox, J. & Mann, M. More than 100,000 detectable peptide species elute in single shotgun proteomics runs but the majority is inaccessible to data-dependent LC-MS/MS. *Journal of proteome research* **10**, 1785-1793, doi:10.1021/pr101060v (2011).

- 2 Danpure, C. J. Variable peroxisomal and mitochondrial targeting of alanine: glyoxylate aminotransferase in mammalian evolution and disease. *Bioessays* **19**, 317-326, doi:10.1002/bies.950190409 (1997).
- 3 Xue, H. H., Sakaguchi, T., Fujie, M., Ogawa, H. & Ichiyama, A. Flux of the L-serine metabolism in rabbit, human, and dog livers. Substantial contributions of both mitochondrial and peroxisomal serine:pyruvate/alanine:glyoxylate aminotransferase. *The Journal of biological chemistry* **274**, 16028-16033 (1999).
- 4 Hieronymus, H. & Silver, P. A. A systems view of mRNP biology. *Genes Dev* **18**, 2845-2860, doi:10.1101/gad.1256904 (2004).
- 5 Shaw, R. J. *et al.* The kinase LKB1 mediates glucose homeostasis in liver and therapeutic effects of metformin. *Science* **310**, 1642-1646, doi:10.1126/science.1120781 (2005).
- 6 Yang, Q., Vijayakumar, A. & Kahn, B. B. Metabolites as regulators of insulin sensitivity and metabolism. *Nat Rev Mol Cell Biol* **19**, 654-672, doi:10.1038/s41580-018-0044-8 (2018).
- 7 Wang, B. *et al.* Suppression of Postprandial Blood Glucose Fluctuations by a Low-Carbohydrate, High-Protein, and High-Omega-3 Diet via Inhibition of Gluconeogenesis. *Int J Mol Sci* **19**, doi:10.3390/ijms19071823 (2018).
- 8 Boudaba, N. *et al.* AMPK Re-Activation Suppresses Hepatic Steatosis but its Downregulation Does Not Promote Fatty Liver Development. *EBioMedicine* **28**, 194-209, doi:10.1016/j.ebiom.2018.01.008 (2018).
- 9 Mihaylova, M. M. *et al.* Class IIa histone deacetylases are hormone-activated regulators of FOXO and mammalian glucose homeostasis. *Cell* **145**, 607-621, doi:10.1016/j.cell.2011.03.043 (2011).
- 10 Just, P. A. *et al.* LKB1 and Notch Pathways Interact and Control Biliary Morphogenesis. *PLoS One* **10**, e0145400, doi:10.1371/journal.pone.0145400 (2015).

REVIEWERS' COMMENTS:

Reviewer #2 (Remarks to the Author):

The authors have added a significant amount of work and substantially improved the manuscript. All major concerns have been addressed adequately.

Reviewer #3 (Remarks to the Author):

The authors have addressed my concerns. They added many new results or comments which clearly improved their manuscript. I have no further suggestions and I recommend the manuscript for publication at Nature communications.

Reviewer #4 (Remarks to the Author):

Title: Lkb1 suppresses amino acid-driven gluconeogenesis in the liver

The manuscript by Just PA and co-authors describes an effort to characterize the role of liver kinase B1, Lkb1, in the regulation of gluconeogenesis in liver.

General comments: The manuscript is very well written, and the authors have utilized a rather smart approach and choice of methods to support their work. I am particularly excited for the proteomics work done in this manuscript. I would firstly like to thank the authors for the time invested to describe their methods and provide all the details. I appreciate the work done with phosphoproteomics, and believe that it greatly contributed to the quality of this manuscript. The results are exciting and promising.

Minor comments:

- 1) The authors succeeded to reveal a totally new role of Lkb1 with proteomics, and I would recommend including and highlighting the word 'proteomics' somewhere in the abstract.
- 2) 'To construct the PPI network in the STRING database, H. sapiens was selected as the organism'. Why H. sapiens and not M. musculus?
- 3) What was the amount of the peptides injected for MS/MS?

Point-by-point response to « Lkb1 suppresses amino acid-driven gluconeogenesis in the liver” by Just PA et al.,

The comments of the reviewer are in black italics, and the response in blue.

First of all, we thank the reviewer for their positive comments.

Reviewer #2 (Remarks to the Author):

The authors have added a significant amount of work and substantially improved the manuscript. All major concerns have been addressed adequately.

Reviewer #3 (Remarks to the Author):

The authors have addressed my concerns. They added many new results or comments which clearly improved their manuscript. I have no further suggestions and I recommend the manuscript for publication at Nature communications.

Reviewer #4 (Remarks to the Author):

Title: Lkb1 suppresses amino acid-driven gluconeogenesis in the liver

The manuscript by Just PA and co-authors describes an effort to characterize the role of liver kinase B1, Lkb1, in the regulation of gluconeogenesis in liver.

General comments: The manuscript is very well written, and the authors have utilized a rather smart approach and choice of methods to support their work. I am particularly excited for the proteomics work done in this manuscript. I would firstly like to thank the authors for the time invested to describe their methods and provide all the details. I appreciate the work done with phosphoproteomics, and believe that it greatly contributed to the quality of this manuscript. The results are exciting and promising.

Minor comments:

1) The authors succeeded to reveal a totally new role of Lkb1 with proteomics, and I would recommend including and highlighting the word ‘proteomics’ somewhere in the abstract.

We thank the reviewer for his comment and have added « proteomic » in the abstract and Introduction section.

2) ‘To construct the PPI network in the STRING database, H. sapiens was selected as the organism’. Why H. sapiens and not M. musculus?

We thank the reviewer for highlighting this error and have done the analysis using the selection *M. musculus* instead of *H. sapiens*.

We obtained similar results, but have modified the Supplementary Figure 5c accordingly.

3) What was the amount of the peptides injected for MS/MS?

Peptides corresponding to one μg of digested proteins was injected for each fraction. This has been added in the Methods section.